# Seasonal enhancement of the viral shunt catalyzes a subsurface oxygen maximum in the Sargasso Sea

Naomi E. Gilbert[1,11,13], Daniel Muratore[2,12,13], Camelia Shopen Gochev [3], Gary R. LeCleir[1], Shelby M. Cagle [1], Helena L. Pound[1], Christine L. Sun[4], Alfonso Carrillo [4], Kimberley S. Ndlovu [4], Ilia Maidanik[3], Ashley R. Coenen[5], Lauren Chittick [4], Jennifer M. DeBruyn [6], Alison Buchan[1], Debbie Lindell [3], Matthew B. Sullivan [4,7], Joshua S. Weitz [8,9,10] ✉ & Steven W. Wilhelm [1] ✉

Subsurface oxygen maxima (SOMs) occur directly beneath the mixed layer of stratified water columns across oligotrophic open ocean basins and have been associated with physical transport processes and localized increases in phytoplankton net primary productivity (NPP). We explore the hypothesis that viral lysis (*i.e.*, the 'viral shunt') increases nutrient recycling and enhances NPP, supporting SOM formation in stratified water columns, focusing on a recurring SOM at the Bermuda Atlantic Time Series (BATS) in the Sargasso Sea. Reanalysis of historical BATS data showed enhanced *Prochlorococcus* and virus-like particle abundances associated with SOMs. Instances of high rates of primary and secondary production observed with oxygen supersaturation further implicate a biological mechanism for SOM formation. Leveraging metatranscriptomes, metaviromes, and polony-based data collected during a Lagrangian cruise (October 2019), we link the viral shunt to SOMs, including evidence of elevated cyanophage abundance and infection of *Prochlorococcus*, and transcriptomic evidence of increased organic matter uptake (*i.e.*, catabolic activity) by copiotrophic bacteria. Cruise data also showed *Prochlorococcus* nitrogen metabolism transcripts consistent with increased responsiveness to bacterial remineralization. These findings illustrate the biogeochemical impacts of enhanced viral lysis in marine systems, including the potential role of the viral shunt in facilitating SOM formation in the oligotrophic oceans.

Marine microbial communities are now facing novel physicochemical environments and potential stresses as a result of changing climate feedback[1]. In turn, shifts in microbially-induced ecosystem function may influence biogeochemical cycles of key elements, such as carbon and oxygen. Quantitative predictions of the feedback between the ocean and atmosphere require incorporating information on microbial processes and their response to physicochemical gradients in the water column. In the oligotrophic open ocean, a significant fraction of atmospheric

carbon dioxide is drawn down via oxygenic photosynthesis by phytoplankton[2]. Moreover, oligotrophic regions such as the Sargasso Sea consistently exhibit subsurface zones harboring dense phototrophic populations[3,4], concurrent with high oxygen concentrations[5,6]. These regions are referred to as "subsurface oxygen maxima" (SOMs).

SOMs have been observed across oligotrophic open ocean regions that include the North Atlantic Subtropical Gyre[5,6], the permanently stratified tropical and subtropical areas of the Atlantic

Ocean[7], the South Pacific Ocean[8], the Mediterranean Sea[9], and sometimes even in large lakes[10]. SOMs can occur at 40–125 m depth, depending on the region and season[11–13], and are formed between the upper mixed layer and the deep chlorophyll maximum[5,7]. The origins of SOMs remain controversial. While some combination of physical and biological mechanisms are likely at work, literature analyzing subsurface oxygen dynamics in the gyres have posed hypotheses ranging from primarily physical (associated with isopycnal subduction) to primary biological (often ascribed to photophysiology). From a physical perspective, highly oxygenated waters may be subducted and remain at high oxygen saturation because of their inability to ventilate[6,9]. From a biological perspective, enhanced primary production from microbial photosynthesis immediately below the mixed layer may generate elevated oxygen autochthonously[8,9,12,13]. At present, there is limited direct exploration of the microbial dynamics that occur at the SOM, despite the fact that microbial activity and interactions fundamentally shape ocean biogeochemistry. Thus, although SOMs are significant and ubiquitous across oligotrophic oceans, key biological mechanisms contributing to SOM formation and maintenance after seasonal stratification remain elusive.

To accurately assess the contribution of microbial interactions to primary production and the physical restructuring of water column depth strata requires high-resolution biological data in a physically well-characterized SOM. One such candidate occurs annually at the Bermuda Atlantic Time-series Study (BATS) long-term oceanographic monitoring site in the Sargasso Sea[5]. Long-term observations at depths commensurate with the BATS SOM reveal elevated *Prochlorococcus* cell abundances, heterotrophic bacterial densities, and particulate organic carbon[3,5]. Virus-like particle abundances (VLPs) have also been reported to reach their maximum levels beneath the mixed layer at BATS during the stratified fall, coinciding with the emergence of the SOM[3]. In theory, a peak in biomass of the dominant microbial populations at the SOM correlated to high VLPs would be consistent with the 'viral shunt' hypothesis, by which viral lysis enhances nutrient recycling and fuels primary production[14–17]. Moreover, historical reports of enhanced microbial and viral population abundances coinciding with the BATS SOM as well as the positive relationships between stratification intensity, SOM formation, and primary/secondary production led us to hypothesize that the BATS SOM harbors a microbial community associated with accelerated dissolved organic matter (DOM) recycling, coupled with enhanced viral lysis. Accelerated DOM recycling and enhanced nutrient retention mediated by viral infection could, in turn, result in oxygen accumulation in an isopycnal below the mixed layer, depending on balance of primary relative to secondary productivity.

Here, we set out to test our hypothesis that SOMs are mediated by an enhanced viral shunt through microbial and biogeochemical characterization of a BATS-proximal SOM sampled during late stratification. In October 2019, over a six-day Lagrangian sampling effort, we collected day/night-resolving metatranscriptomes along with direct measurements of population-specific densities of the dominant taxa from depth profiles sampling above, at, and beneath the SOM. We quantified cyanophage abundances and infections in the SOM using the polony and iPolony methods[18,19] and community viromic assessment[20], focusing on cyanophages capable of infecting *Prochlorococcus*. Our results, combining high-resolution temporal analysis with historical data from BATS, suggest that enhanced viral lysis leads to tight feedback between photoautotrophs and heterotrophs and the repeated, seasonal emergence of SOMs in late fall associated with ocean stratification in an archetypal oligotrophic gyre.

## Results and discussion

### Seasonal stratification is accompanied with persistent oxygen saturation and particulate matter within the sub-mixed layer

Depth profiles were collected every 4 h aboard the RV *Atlantic Explorer* following a Lagrangian cruise track starting at BATS (31° 40′ N, 64°10′ W)

on October 12th, 2019 and ending on October 17th, 2019 (Fig. 1a and b). Density profiles derived from conductivity, temperature, density (CTD) measurements indicated a highly stratified upper water column with a stable mixed-layer depth around 50 m for the duration of the sampling period (Fig. 1c). A stable subsurface peak in oxygen saturation was apparent in the isopycnal about 5 m beneath the mixed layer depth (Fig. 1d). This layer, which we refer to as the "SOM", was situated approximately 50 m above the deep chlorophyll maximum (DCM; Fig. 1d and e, Supplementary Fig. 1). The SOM also displayed the highest beam attenuation values, on average 1.52-fold higher than the DCM (sd ± 0.14), and 1.09-fold higher than the upper mixed layer (sd ± 0.04), suggesting peak concentrations of particulate organic matter (POM) in the 0.5–20 µm size range[21,22] (Fig. 1f and Supplementary Fig. 1). CTD profiles measured every 4 h reveal a diel cycle in beam attenuation within the SOM (RAIN nonparametric test, $p < 1e-10$), suggesting these particles had growth/decay cycles and were not solely allochthonous detritus (Fig. 1f). These observations are consistent with a previous analysis conducted at BATS which found enhanced colored dissolved organic matter (cDOM) levels associated with elevated bacterial abundance and production in and around what we identify as the SOM layer[5]. The SOM during our study had a mean increase of 22.14 (sd ± 4.20) µmol $O_2$/kg (11.57% increase) compared to the upper mixed layer, and a mean increase of 17.43 (sd ± 5.66) µmol $O_2$/kg (8.92% increase) compared to the DCM. These differences result in an average increase in oxygen saturation of 6.47% (sd ± 1.22%) in the SOM from the upper mixed layer, and an increase in oxygen saturation of 13.9% (sd ± 1.93%) in the SOM compared to the DCM (Fig. 1d). The chlorophyll fluorescence in the SOM, however, was only 26.9% (sd ± 4.47%) that of the DCM (Fig. 1e).

We reanalyzed oceanographic climatological data collected from 1988 to 2019 to contextualize our October 2019 cruise data within the physical and biogeochemical context of the BATS SOM. Prior work on sub-mixed layer oxygen dynamics in the BATS region used profiling float data taken across four seasonal cycles from 2010 to 2013 and the BATS chemical measurements of organic matter, oxygen, and nitrate to argue that isopycnal subduction could explain oxygen accumulation from 50 to 100 m during the stratified season, despite drawdowns via heterotrophic activity[6]. In our analysis, we focus on the 10 m below the mixed layer depth and expand the record for analysis by using a standardized method of sensor data processing and thermodynamic conversions, calibrated against bottle oxygen measurements using the Winkler method (see Methods) – resulting in 5072 CTD casts with high-quality oxygen profiles spanning 31 years. We used these profiles to characterize the change in mean oxygen saturation in the 10 m beneath the mixed layer depth as a function of the time of year (Fig. 2b). Nonlinear least squares fitting of a sinusoidal regression identified an annual fluctuation in sub-mixed layer oxygen saturation of approximately 6.79% (6.71, 6.88; 95% CI, $p < 1e-10$), from a baseline of 100.50% (100.42,100.59; 95% CI, $p < 1e-10$), with an annual minimum in mid-February (decimal year=0.123, 0.121,0.125; 95% CI, $p < 1e-10$). Due to the symmetry of the sine function, this sinusoidal regression indicated a maximum in August (Fig. 3). These results suggest that on average, over the BATS climatology, the annual SOM's intensity reaches its maximum in August. The maximum in sub-mixed layer oxygen saturation builds after the deep mixing period ends (typically in March–April) to reach a maximum in August, until it dissipates at the onset of the next deep mixing event in winter (Fig. 2a). By defining the SOM in reference to the mixed layer depth ascertained through a density surface-based criterion[23], we adjust for year-to-year variation in the depth placement of specific isopycnals. Isopycnal-specific analyses of the isopycnal surfaces that lie directly beneath the mixed layer (in the range of $\sigma\theta$= 24.4-24.6) show seasonal dynamics in oxygen saturation and total concentration with two peaks – one during the deep mixing in March (~95%), and another in August (~108%), suggesting oxygen accumulation post-stratification in the narrow band of water directly beneath the mixed layer (Fig. 3).

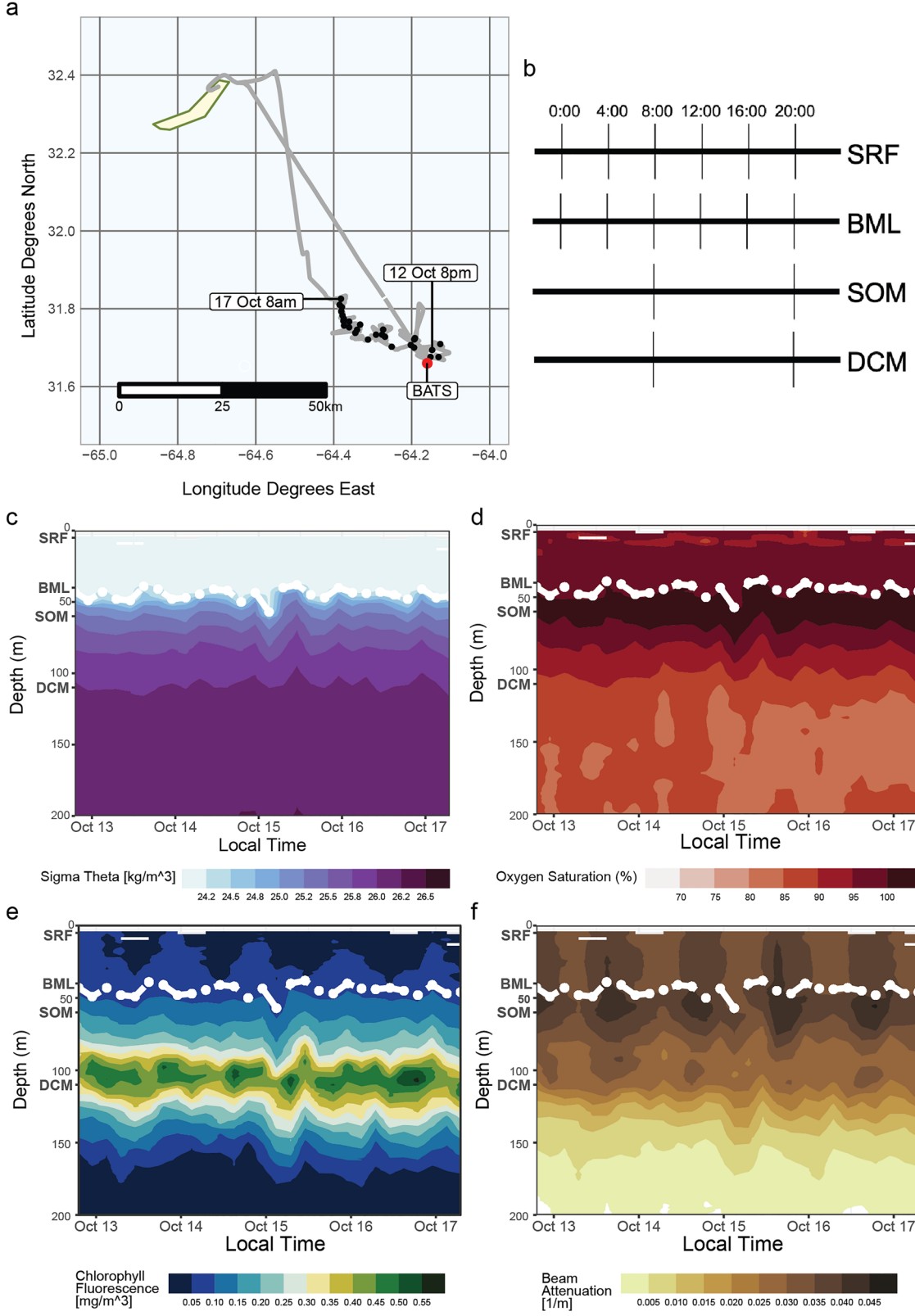

**Fig. 1 | Depth resolved diel sampling during the 2019 fall cruise at BATS.**
**a** Lagrangian sampling track starting at the BATS station (red point). Each black point represents a CTD-cast starting Oct. 12th, 2019, at 20:00 local time (GMT-3) and ending on Oct. 17th, 2019, at 08:00. **b** Schematic of a single day in the time series sampling profile. The surface (SRF) and base of the mixed layer (BML) were sampled every four hours, while the subsurface oxygen maximum (SOM) and deep chlorophyll maximum (DCM) were sampled every twelve hours. Panels c-f show

CTD profiles interpolated over the entire time series (taken every four hours from the surface to 200 m depth) for **c** potential density (sigma-theta) (kg/m$^3$) **d** Oxygen saturation (%) **e** Chlorophyll fluorescence (mg/m$^3$) and **f** Beam attenuation (1/m); with the mixed layer depth (as defined by 0.125 difference in sigma-theta from 10 m) indicated in white. Depth names are designated at their approximate location in bold.

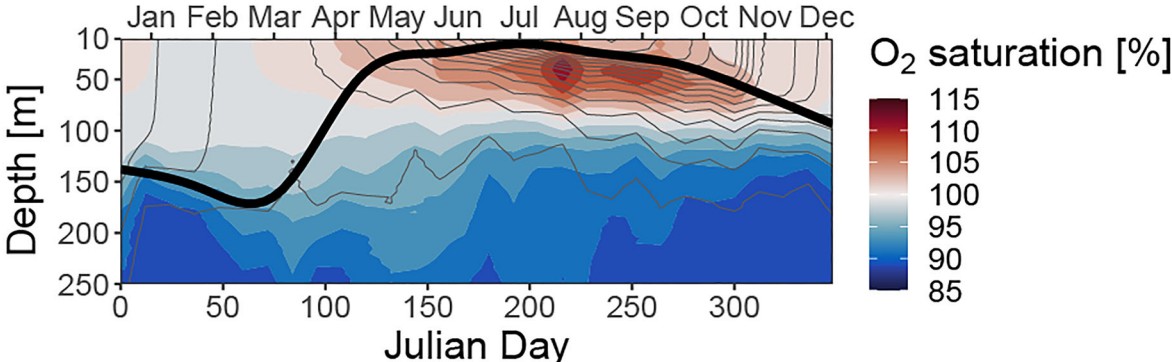

We also reanalyzed primary production and bacterial production experiments conducted in situ over the course of the BATS climatology with a focus on the subset of these experiments that take place in the SOM (approximately 7 of 112). We find that bacterial secondary production and carbon fixation positively correlated with oxygen saturation in production experiments conducted at 40 m-60 m (Supplementary Fig. 2). Furthermore, the experiments that have the highest oxygen saturation and most closely resemble a SOM also have the highest primary and secondary production rates in the BATS climatology (Supplementary Fig. 2). Our findings of increased oxygen saturation in the narrow density surface directly beneath the mixed layer throughout the stratified period, as well as observations of elevated primary and secondary production rates measured in this narrow feature, suggest that biological activity in the SOM – and not

**Fig. 2 | Interannual mean oxygen saturation profiles from the BATS CTD record from 1996-2019. a** Oxygen saturation data are binned by 5 m depth windows and 12-day means throughout the year. The black line indicates interannual average mixed layer depth for a 10-day window as determined by 0.125 kg/m³ change in potential density from a reference pressure of 10 db. The color bar is scaled such that white refers to an oxygen saturation state of 100% (in balance with the atmosphere), blue colors are undersaturated (suggesting respiration), and red colors are supersaturated (suggesting positive net primary production). Gray lines indicate the annual depth trajectory of potential density surfaces binned at 0.1 kg/m³ resolution, such that color changes between gray lines are along-isopycnal changes in oxygen saturation (with changes from white to red indicating oxygen production and changes from white to blue indicating respiration). Data were retrieved from http://bats.bios.edu/bats-data/ **b** Annual BATS oxygen profiles by month of year. Data are binned into 2 m depth bins and color indicates smoothed oxygen concentration as determined by CTD optodes. Black line indicates mixed-layer depth as determined by a 0.125 difference in sigma-theta from a reference pressure of 10 db. White bars indicate missing data for that month.

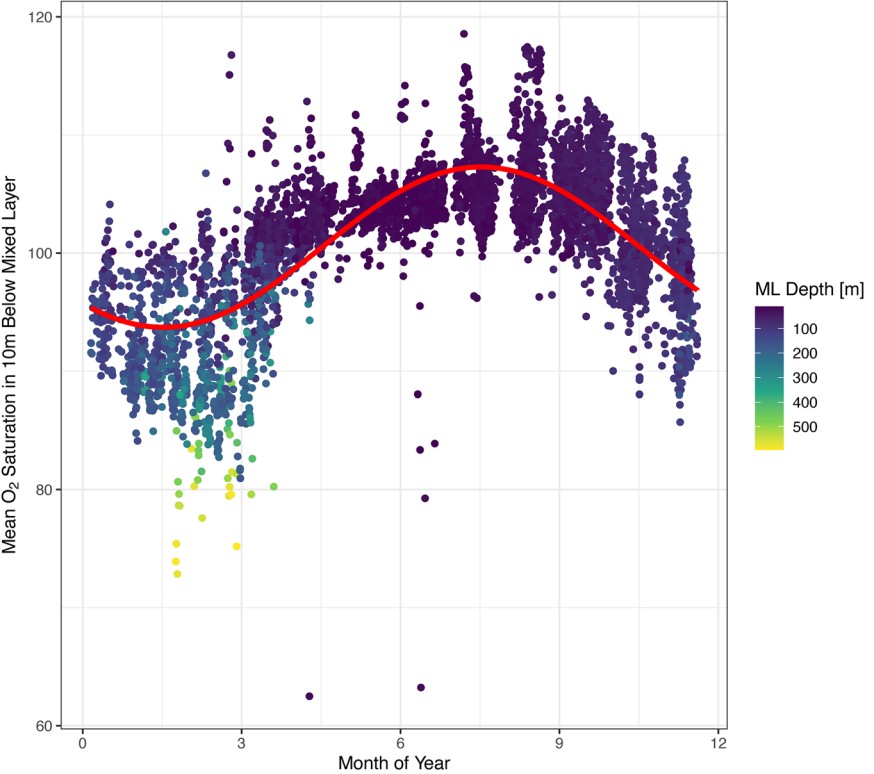

**Fig. 3 | Seasonal cycle in sub-mixed layer oxygen dynamics at BATS.** Each point represents the mean oxygen saturation percentage for one of 5054 casts from BATS cruises where optode data matched Winkler method bottle calibrations (see Methods). The x-axis is the month of the year in which the BATS cruise took place, the y-axis is the mean oxygen saturation between the mixed layer depth and 10 m below the mixed layer depth, with mixed layer depth determined by a change in potential density of 0.125 kg/m³ from a reference pressure of 10 db. Point color indicates the calculated mixed layer depth of that cast. A nonlinear least squared fit of a sinusoidal regression was calculated and is shown as the red line. The model specification is: $\mathrm{mean}\left(O_{2\mathrm{saturation}}\right) = a + b sin\left(2\pi*\left(c + \mathrm{mod}\left(\frac{\mathrm{decimalyear}}{12}\right)\right)\right)$. Decimal year goes from 0 (Jan 1) to 1 (Dec 31). Estimated parameters with standard errors and p-values are: $a = 1.020e + 02$ std.error: $4.559e-02$ $p < 2e-16$, $b = -2.993e + 00$ std.error: $6.536e-02$ $p < 2e-16$, $c = 5.924e-01$ std.error: $3.376e-03$ $p < 2e-16$.

physical processes alone – drive oxygen accumulation. These processes are distinct from activity in the surface mixed layer and underlying subphotic waters throughout the stratified period.

### SOM microbial community activity is more similar to the mixed layer than the deep chlorophyll maximum

We collected metatranscriptomes from depth profiles every 12 h (day = 08:00, night = 20:00) during our Lagrangian sampling effort in October 2019 (Fig. 1b). These samples allowed us to compare the expression of 17,798,931 genes across eukaryotic plankton, bacteria, and viruses between the upper mixed layer, SOM, and the DCM. We constructed a PCA ordination of metatranscriptome gene expression profiles (Supplementary Fig. 3). The axis explaining most of the variation between metatranscriptomes (54.67%) separated the DCM samples from the other depths, meaning that SOM metatranscriptomes more closely resemble samples from the surface and the base of the mixed layer than the DCM (Supplementary Fig. 3). The second PCA axis, explaining 19.86% of total variance, established a gradient from surface samples (highest PC2 values) to the base of the

mixed layer to SOM samples (lowest PC2 values, Supplementary Fig. 3). Metatranscriptomic profiles suggest the SOM microbial community transcription resembled the surface/base of the mixed layer more closely than it does the DCM, with some divergence in community-level expression profiles.

Next, we evaluated to what extent differences in community expression were due to difference in microbial community composition across these depth layers versus differences in the relative expression of different functional genes among shared taxa. We applied hierarchical clustering to taxon-specific expression of *rpoB* and *RPB1*, a pair of core conserved housekeeping genes (RNA polymerase subunit B) spanning Archaea (*rpoB*), Bacteria (*rpoB*), and Eukaryota (*RPB1*) (Fig. 4a). The taxonomic distribution of *rpoB/RPB1* transcripts in the SOM more closely resembled that of the surface and base of the mixed layer than the DCM (Fig. 4a and b). The surface/base of the mixed layer and SOM samples were characterized by high relative *rpoB* transcript abundances of the High-Light II *Prochlorococcus* ecotypes, and *rpoB* transcripts assigned to the orders Flavobacteriales and Rhodospirillales (Fig. 4a). The SOM samples had

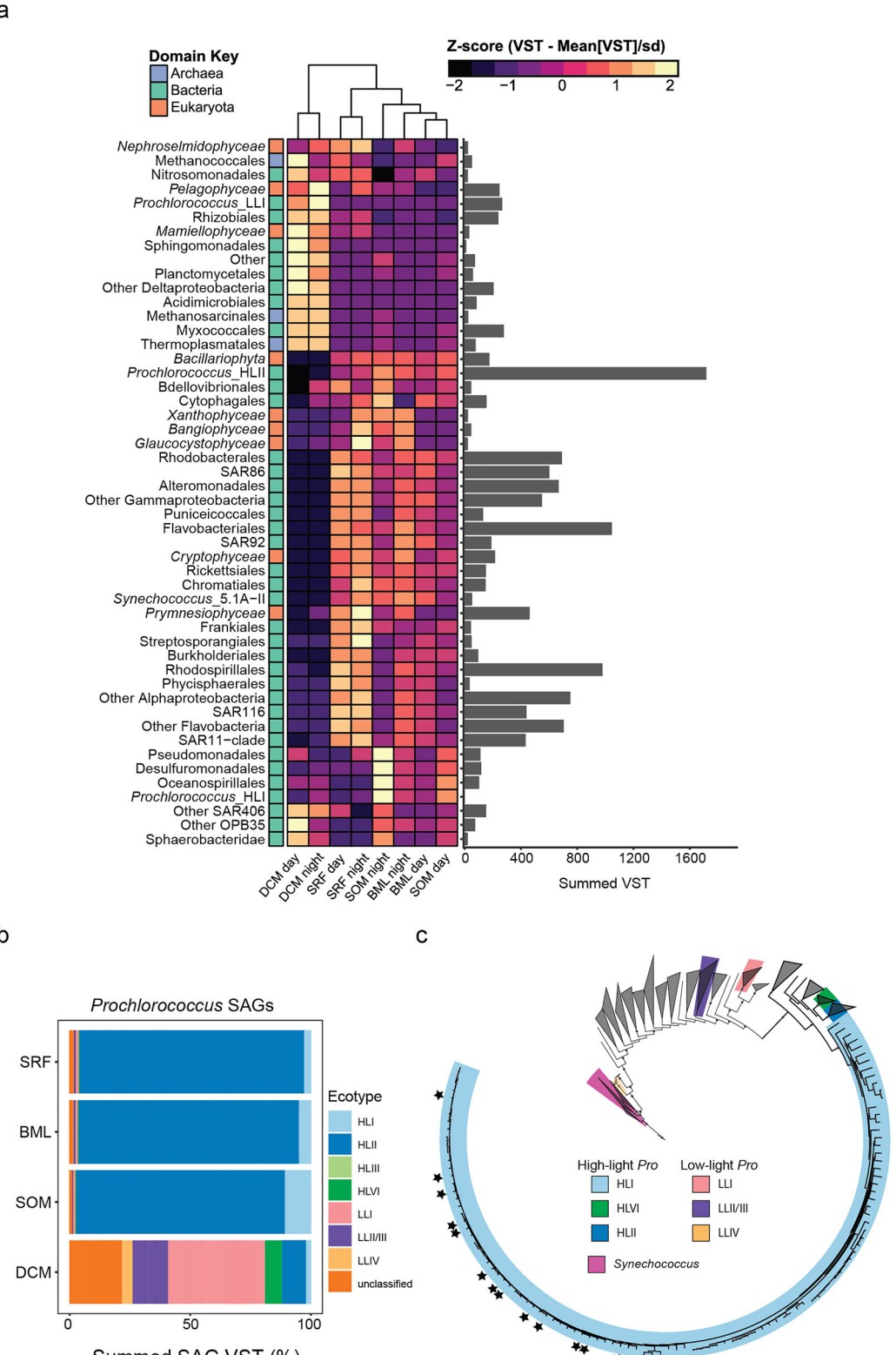

a

b

*Prochlorococcus* SAGs

c

High-light *Pro*     Low-light *Pro*

elevated relative *rpoB* transcript abundances for the Protobacteria orders Pseudomonadales, Oceanospirillales, and Desulfuromonadales, the Bacteriodetes order Cytophagales, and *Prochlorococcus* High-Light I ecotypes relative to the surface/base of the mixed layer samples (Fig. 4). A peak in *Prochlorococcus* High-Light I ecotypes has been observed in a SOM in the Mediterranean Sea[24,25]. Statistical assessment of contig-level *rpoB* and *RPB1* abundance differences with depth also

identified significantly (BH-adjusted $p < 0.1$) elevated transcript abundances in the SOM of contigs with *rpoBs* assigned to known copiotrophic taxa (e.g., Rhodobacterales, Flavobacteriales, Oceanospirillales, Cytophagales; Supplementary Fig. 4). These taxa were associated with environments analogous to phytoplankton blooms, for example, where algal-derived organic matter primarily supplies these rapid DOM recyclers[26–29]. A seasonal sub-mixed-layer

**Fig. 4 | The SOM resembles the upper mixed layer, with key differences in taxonomic composition. a** Hierarchically clustered heatmap of *rpoB/RPB1* normalized transcripts (variance stabilizing transformation, VST) averaged across depth/time collected (day = 8 am, night = 8 pm) and standardized by row (Z-score, ([Observed VST – Mean VST]/standard deviation). VST values are summed across order-level taxonomy for prokaryotes, except for *Prochlorococcus* and *Synechococcus*, which are summed to the ecotype-level. VST values are summed across class-level taxonomy for eukaryotes. Each row is color-coded by domain-level taxonomic information. The bar plot to the left shows the total VST for each row across all samples. **b** Stacked bar plot showing the distribution of transcripts assigned to single-celled amplified genomes (SAGS) from different *Prochlorococcus* ecotypes across depth. **c** Maximum Likelihood phylogeny of cyanobacterial genomes. *Prochlorococcus* ecotypes and *Synechococcus* genomes detected in the metatranscriptomes are color coded. Stars indicate SAGs that had significantly (BH-adjusted p ≤ 0.1) highest *rpoB* transcript abundances at the SOM. Because these all fell into the HLI clade, the other clades are collapsed for visual purposes. SAGs and cyanobacterial genomes are from Berube et al.[77]. Depth layers sampled are surface, defined as 5 m (SRF), base of mixed layer (BML) defined by a change in potential density of 0.125 kg/m³ from a reference pressure of 10 db, the subsurface oxygen maximum (SOM) depth, and the deep chlorophyll maximum (DCM).

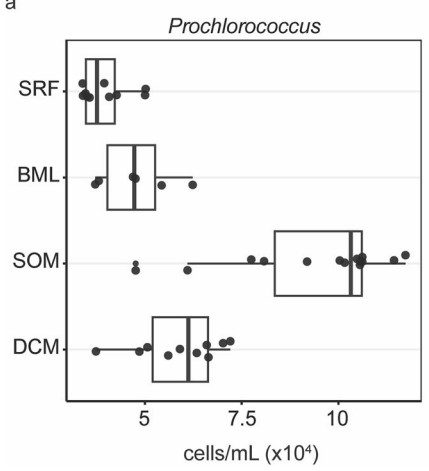

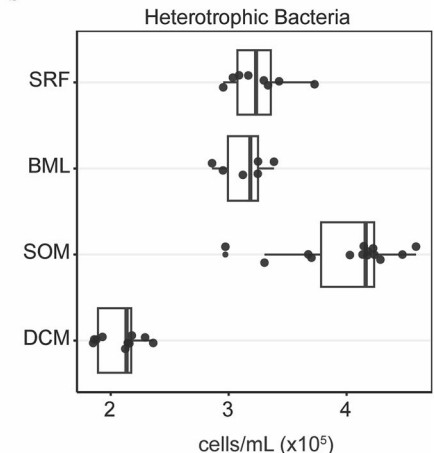

**Fig. 5 | Bulk cellular abundances of *Prochlorococcus* and heterotrophic bacteria.** Each circle is an independent water sample (a biological replicate) collected at 8 am or 8 pm at each of the 4 depths across the cruise (SRF *n* = 10, BML *n* = 6, SOM *n* = 14, DCM *n* = 10), and the abundances were measured using flow cytometry. Depth layers sampled are surface, defined as 5 m (SRF), base of mixed layer (BML) defined by a change in potential density of 0.125 kg/m³ from a reference pressure of 10 db, the subsurface oxygen maximum (SOM) depth, and the deep chlorophyll maximum (DCM). **a** Abundance of *Prochlorococcus* cells. **b** Abundance of heterotrophic bacterial cells. In the boxplots, the center line indicates the median, the box edges represent the 25th (lower, quartile 1) and 75th (upper, quartile 3) percentiles, and the whiskers extend to the smallest and largest values within 1.5 times the interquartile range below quartile 1 and above quartile 3, respectively.

*Rhodobacteraceae* maximum has also been previously observed at BATS[3]. Conversely, we observe genome "streamlined" heterotrophs such as SAR116, SAR86, and SAR92[30], that are primarily active in the surface/base of the mixed layer relative to the SOM (Fig. 4a).

We then compared the changes in population structure as inferred through the housekeeping gene analysis to depth profiles of cell counts as determined by flow cytometry. *Prochlorococcus*, which had increased transcriptional abundance in the SOM, also had increased cell abundances compared to the surface (BH-adjusted *p* < 0.0001), base of the mixed layer (BH-adjusted *p* = 0.0012) and DCM (BH-adjusted *p* = 0.0229; Fig. 5). Peaks in *Prochlorococcus* high-light ecotype abundances have been observed below the mixed layer at BATS and the North Pacific Subtropical Gyre[3,4]. Heterotrophic bacterial cell abundances were also elevated at the SOM compared to the surface (BH-adjusted *p* = 0.0351), base of the mixed layer (BH-adjusted *p* = 0.0165), and DCM (BH-adjusted *p* < 0.0001; Fig. 5). Oxygen saturation above 100% in this isopycnal can result from greater primary production ($O_2$ supply) compared to respiration ($O_2$ demand). While both primary producers and heterotrophs had overall increased abundances in the SOM, the relative increase of *Prochlorococcus* was higher, with a mean heterotrophic bacterium-to-*Prochlorococcus* ratio of 4.4 as opposed to 8.2 in the mixed layer (Supplementary Data 1). This suggests that oxygen supersaturation in the SOM could arise from disproportionate increases in *Prochlorococcus* (driving increased primary production) relative to heterotrophic bacteria.

## Taxa shared between the mixed layer and SOM are enriched in transporter transcripts related to nitrogen recycling

The joint analysis of expression and composition is consistent with the hypothesis that the SOM supports a microbial community containing similar taxa to the surface and base of the mixed layer, but in an alternative functional state maintaining higher population densities and/or activities of heterotrophic bacteria and *Prochlorococcus*. To further explore this hypothesis we compared functional gene expression profiles of the surface, base of the mixed layer, SOM, and DCM with gene enrichment analyses based on order-level (for heterotrophic bacteria), class-level (Eukaryotes), or genus-level (for Cyanobacteria, combining all *Prochlorococcus* ecotypes) aggregates of functional genes annotated via KEGG orthology [KO[31]]. Using ratios of KO transcript abundances to the corresponding *rpoB/RPB1* ("*rpoB* ratio"; see Methods), we identified 604 KO-annotated genes that had either significantly (BH-adjusted *p* < 0.1) higher (193) or lower (411) transcripts relative to that taxon's *rpoB/RPB1* expression at the SOM compared to all other measured depths, spanning eukaryotic and prokaryotic taxa (Supplementary Fig. 5 and Supplementary Data 2).

Out of the 604 KO-annotated genes, 25 were assigned to *Prochlorococcus* (Supplementary Fig. 6a and Supplementary Data 2). Three of these had significantly higher transcript abundances at the SOM relative to all other depths (Supplementary Fig. 6a and Supplementary Data 2). These are *mrp* (ATP-binding protein involved in chromosome partitioning; K03593), *rpmG* (large subunit ribosomal protein L33; K02913), and *amt* (ammonium transporter, K03320; Supplementary Fig. 6a and Supplementary Data 2). The *Prochlorococcus* ammonium transporter, *amt*, had the highest *rpoB* ratio at the SOM (-2.5 *amt:rpoB*) compared to all other *Prochlorococcus* KOs. *Prochlorococcus amt* expression also increased at dusk in the mixed layer, but not at the DCM (Supplementary Fig. 6b).

Seventy-four KO-annotated genes belonging to the KEGG "transport" category assigned to eukaryotes, heterotrophic bacteria, and

cyanobacteria were identified that were significantly (BH-adjusted $p < 0.1$) higher or lower transcript values at the SOM compared to all other depths measured (Supplementary Data 2 and Supplementary Data 3). Transporter transcripts for inorganic nutrients such as iron/ other metals, phosphate, and nitrate were lowest at the SOM (Supplementary Data 3). Heterotrophic bacterial transcripts were significantly highest at the SOM included transporters for (nitrogen-containing) organic substrates, including polar/branched-chain amino acids, oligopeptides, and polyamines (Supplementary Data 3). While expression cannot directly be substituted for biochemical activity[32], we identified higher transcription related to uptake of organic nitrogen-containing molecules among heterotrophs and enhanced ammonium uptake by *Prochlorococcus* (the latter of which is mentioned above), alongside decreased expression of transporters for other nutrients by all taxonomic groups. This result is consistent with increased uptake of organic nitrogen-containing molecules among heterotrophs and ammonium uptake among *Prochlorococcus* at the SOM.

The enhanced *amt* expression by *Prochlorococcus* at the SOM could be due to different phenomena. Increased transcript levels could be a response to nitrogen starvation[33] due to rapid drawdown of nitrogen sources (including ammonium) by competing cells. Alternatively, this could be a response to enhanced ammonium availability. While *amt* transcription has been shown to be constitutively high for some *Prochlorococcus* strains in the HLI and LLIV ecotypes[33–35], the addition of ammonium after severe nitrogen starvation resulted in an increase in transcript levels[34]. Ammonium is not part of the standard suite of inorganic nutrients measured via monthly cruise expeditions at BATS, so historical observations of ammonium concentrations in the SOM are sparse[5,36]. Further, because the realized concentration of nutrients in situ is influenced by input, production, drawdown, and abiotic transformation, extrapolating realized nutrient availability from measured nutrient concentrations remains challenging. Since *ntcA* transcription is low in the presence of sufficient ammonium[33,34], the combined finding of significantly lower *ntcA* transcript levels (Supplementary Fig. 6a) and significantly higher *amt* transcript levels normalized to *rpoB* transcripts, suggests a higher flux of ammonium and a role for this nitrogen acquisition pathway for *Prochlorococcus* at the SOM.

We propose that the enhanced *amt* expression at the SOM is likely due to a combination of physicochemical and biological factors, where *Prochlorococcus* satisfies its nitrogen demand under competition via rapid ammonium assimilation, as suggested in previous studies[34,37]. Vertical injection of ammonium towards the SOM from depth is possible, however enrichment in *amt* transcripts is concentrated at the SOM and not at the DCM, suggesting a different supply mechanism. *Prochlorococcus amt* transcript levels were higher at dusk for both the SOM and surface/base of the mixed layer (Supplementary Fig. 6b). Phototrophic *amt* expression has been shown to occur mainly at dusk, and was linked to enhanced ribosome and protein synthesis expression at night[38]. These patterns are also consistent with diel expression of *Prochlorococcus amt* in laboratory experiments[35]. The transport of organic compounds by heterotrophic bacteria at the SOM likely satisfies their carbon and nitrogen demand under competition, as copiotrophic bacteria are suited to use diverse inorganic and organic resources for growth[29]. Collectively, these observations relate to each other in the context of the microbial loop – heterotrophic bacteria catabolize DOM, excreting ammonium as a byproduct, fueling carbon fixation by primary producers, who would then release DOM[39,40]. In this case, we hypothesize that heterotrophic degradation of organic matter provided additional ammonium that supported *Prochlorococcus* growth. Our hypothesis led us to ask - what was the source of DOM for heterotrophs?

## Elevated signatures of viral infection of prokaryotes in the SOM
Virus-mediated mortality of *Prochlorococcus* is hypothesized to be a significant contributor to the release of DOM into the environment[14,15,41]. Virus-induced lysis and release of DOM is challenging to estimate directly[19], hence proxies for the strength of potential viral lysis are often used, including viral abundance, the percentage of infected cells when possible, and intracellular viral transcriptional activity. Here, we quantified cyanophage abundance in the water column using the polony method[18] and found that, on-average, T4-like cyanophages were more abundant in the SOM than in the mixed layer by 3.12-fold (BH-adjusted $p = 0.0163$) and 5.54-fold (BH-adjusted $p = 0.0007$) relative to the surface and the base of the mixed layer respectively (Fig. 6a, Supplementary Data 1). Likewise, via the iPolony method[19], we found that the total number of infected cells was higher in the SOM by 11.70-fold (BH-adjusted $p < 0.0001$) and 3.44-fold (BH-adjusted $p = 0.014$) than in the surface and base of the mixed layer respectively (Fig. 6b, Supplementary Data 1). This equates to roughly 2.8% of *Prochlorococcus* cells infected by T4-like cyanophages at the SOM (Supplementary Fig. 7).

Additionally, we detected higher free T7-like cyanophages in the SOM by 19.62-fold (BH-adjusted $p < 0.0001$), 15.42-fold (BH-adjusted $p = 0.0002$), and 2.79-fold (BH-adjusted $p = 0.1524$) relative to the surface, base of the mixed layer, and DCM respectively (Fig. 6c, Supplementary Data 1). The total number of T7-like cyanophage-infected *Prochlorococcus* cells was also higher in the SOM by 11.26-fold (BH-adjusted $p < 0.0001$) and 5.42-fold (BH-adjusted $p = 0.0057$) relative to the surface and base of the mixed layer respectively (Fig. 6d, Supplementary Data 1). This equates to ~4.1% of the population of *Prochlorococcus* being infected by T7-like cyanophages at the SOM (Supplementary Fig. 7). Together, ~7% of *Prochlorococcus* was infected by cyanophages in the SOM.

We also assessed the intracellular expression of assembled virus Operational Taxonomic Units (vOTUs) from collected viromes, comparing their expression values across depth. First, we assigned vOTUs to T4- and T7-like phage groups broadly infecting prokaryotes based on the presence of hallmark genes and found that a wide range of T4- and T7-like vOTUs had the highest expression values at the SOM compared to the mixed layer (Supplementary Fig. 8, see Methods for hallmark gene details). Only a smaller subset of mostly T7-like vOTUs had expression values localized to the DCM (Supplementary Fig. 8). Next, using phylogenies of translated hallmark genes for T4-like (Gp23) and T7-like (DNA Pol A) phages, we assessed putative vOTUs related to isolated cyanophages infecting *Prochlorococcus* and *Synechococcus* hosts (Supplementary Figs. 9 and 10). Putative phage scaffolds related to T4-like cyanophage isolates (Supplementary Fig. 9) had 0.7–3.76 and 0.96–11.02 $\log_2$ fold-change more transcripts at the SOM compared to the surface and the DCM, respectively (Supplementary Data 4). Putative phage scaffolds related to T7-like cyanophage isolates (Supplementary Fig. 10) had 2.06–6.23 and 1.43–6.23 $\log_2$ fold-change more transcripts at the SOM compared to the surface and the DCM (Supplementary Data 4). Although T7-like cyanophage abundance was highest at the SOM, the total number of T7-cyanophage infected cells at the SOM was similar to the DCM (Fig. 6d, Supplementary Data 1). We also found transcripts for other phages related to viral isolates known to infect heterotrophic bacteria that were significantly elevated at the SOM (Supplementary Figs. 9 and 10), suggesting that enhanced viral replication may be expanded to bacterial taxa beyond cyanobacteria at the SOM.

The combined results indicate elevated viral infection and production of *Prochlorococcus* cyanophages as well as elevated transcript abundances of heterotrophic phage in the SOM. Previous analysis found increases in virus-like particles (VLPs), representing the total dsDNA virioplankton particle concentration, in the SOM over a period of 10 years at BATS concomitant with increases in *Prochlorococcus* and bulk-*Rhodobacteraceae* cell abundances[3] (see Supplementary Fig. 11 for reanalysis of these data with BATS oxygen profile data incorporated). Further, peak T4 and T7-like cyanophage abundances (inferred via cellular metagenomes) have been documented below the mixed

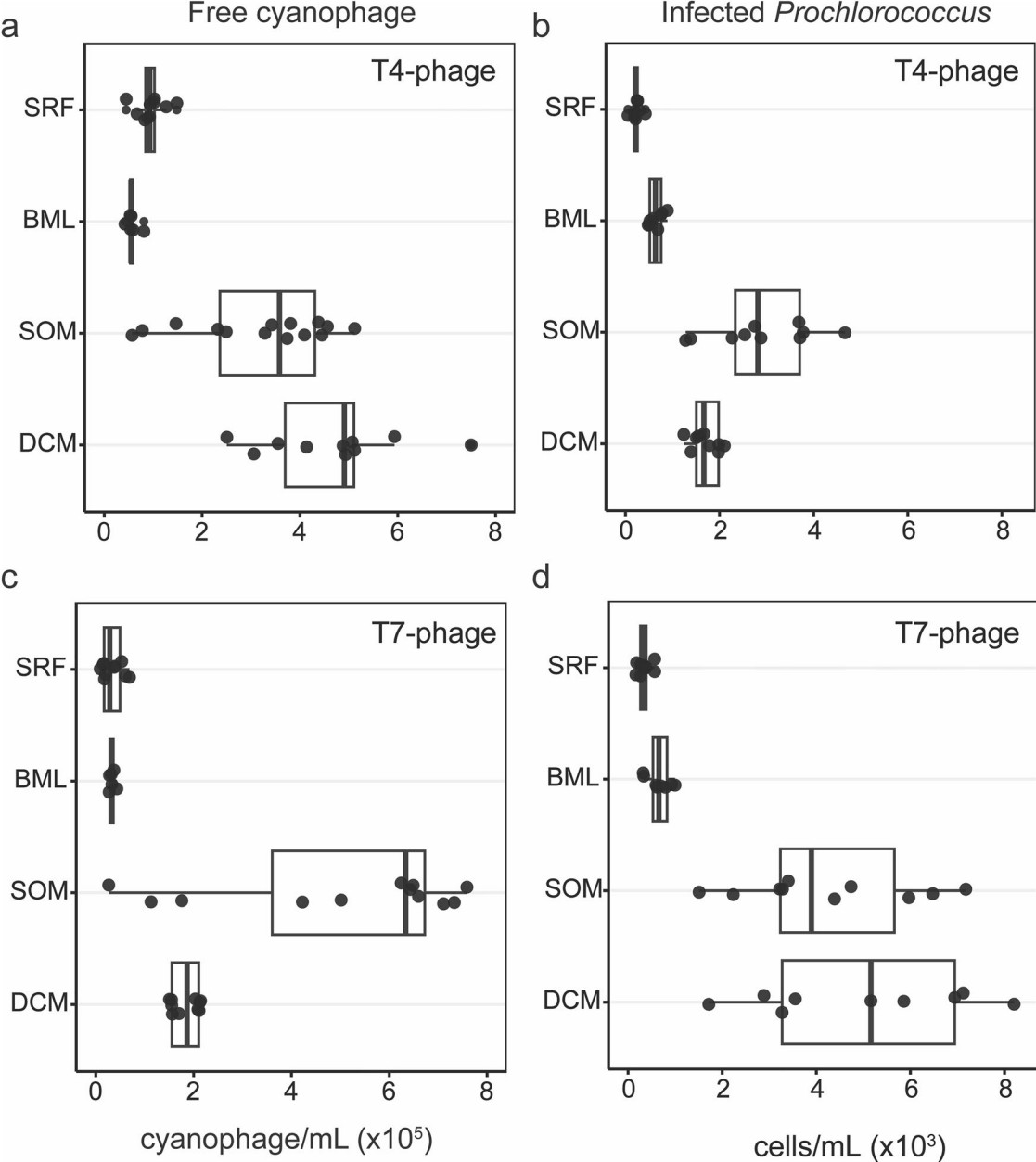

**Fig. 6 | T4- and T7-like cyanophages targeting *Prochlorococcus* are abundant in the extracellular fraction and actively infecting at the SOM.** Depth layers sampled are surface, defined as 5 m (SRF), base of mixed layer (BML) defined by a change in potential density of 0.125 kg/m³ from a reference pressure of 10 db, the subsurface oxygen maximum (SOM) depth, and the deep chlorophyll maximum (DCM). **a**) Abundance of free T4-like cyanophages and **b** Number of *Prochlorococcus* cells infected by T4-like cyanophages, across depth. **c** Abundance of free T7-like cyanophages and **d** Number of *Prochlorococcus* cells infected by T7-like cyanophage across depth. Free cyanophages were quantified by the Polony method which measures phage DNA in the <0.2 μm fraction, while the number of infected cells was determined by the iPolony method which quantifies the number of *Prochlorococcus* with intracellular phage DNA. See Supplementary Fig. 8 for the percent of *Prochlorococcus* infected by T4-like and T7-like cyanophages. Each circle is an independent water sample (a biological replicate) collected at 8 am or 8 pm at each of the 4 depths across the cruise (for polony data: SRF $n = 10$, BML $n = 6$, SOM $n = 14$, DCM $n = 10$; for iPolony data: SRF $n = 10$, BML $n = 6$, SOM $n = 10$, DCM $n = 10$). In the boxplots, the center line indicates the median, the box edges represent the 25th (lower, quartile 1) and 75th (upper, quartile 3) percentiles, and the whiskers extend to the smallest and largest values within 1.5 times the interquartile range below quartile 1 and above quartile 3, respectively.

layer across ocean basins[42]. The elevated standing stock of viral particles and evidence of heightened levels of viral infection during our Lagrangian cruise suggest that a large population of extracellular virions are maintained by infection and replication at the SOM. To date, all known *Prochlorococcus* phages are obligately lytic[43,44], and lytic cyanophage have been well documented at BATS[45]. Indeed, the position of the SOM in the water column may favor lytic reproduction of viruses, as light levels are attenuated at this depth, reducing the likelihood of viral particle inactivation via UV-induced damage[46].

Additionally, enhanced nutrient availability supplied to the host could play a role in promoting lytic infection at the SOM (reviewed in ref. 47). Altogether, these results suggest the SOM harbors increased viral infection across multiple bacterial taxa, ranging from cyanobacterial to heterotrophic bacterial hosts.

### Toward a theory of an enhanced viral shunt in the SOM
In this paper, we explored potential ecological mechanisms underlying the recurrent feature of a SOM in the Sargasso Sea. We did so by

assessing both cellular and viral activity proxies of the current BATS Lagrangian dataset along with detailed historical abundance data. Through our analysis of historical BATS data to trace the emergence of the SOM over the past 30 years, we found corroborating evidence for a relationship between the SOM, primary production, *Prochlorococcus* populations, and virioplankton abundances. By contextualizing our Lagrangian cruise study with long-term dynamics, we propose that the SOM contains a microbial community with enhanced net primary production, accelerated by a seasonal viral shunt, wherein enhanced viral infection of primary producers stimulates heterotrophic organic matter remineralization, providing inorganic nutrients to fuel primary production and oxygen accumulation.

Previous studies have provided evidence that SOMs are associated with enhanced DOM levels in the form of cDOM, which has been shown to stimulate the growth of heterotrophic bacteria[48]. Other studies have shown bioavailable cDOM is enriched in the lysate of *Prochlorococcus* after infection by cyanophage[49]. Metabolomic analysis of marine bacterial lysates has also been shown to be enriched in amino acids and other organic nitrogen substrates upon infection by *Synechococcus* phage[50] and heterotrophic bacteriophages[51–56]. Catabolism of organic matter by heterotrophic bacteria produces abundant inorganic nitrogen in the form of ammonium, which *Prochlorococcus* competitively sequesters to satisfy its nitrogen demand[37]. Indeed, laboratory studies have demonstrated that DOM enrichment, driven by viral lysis, supported phytoplankton growth by supplying excess ammonium[17,57]. Additionally, the increased dusk *Prochlorococcus amt* expression coinciding with enhanced beam attenuation at dusk suggest a potential coupling of these processes at the SOM. Indeed, the SOM in the stratified season likely supports optimal lytic viral replication, where light is sufficient to allow phototrophic metabolism, yet attenuated to prevent viral particle degradation. Additionally, our transcriptomics data suggest alleviated nutrient limitation below the mixed layer, which likely contributes to enhanced microbial abundances and viral reproduction.

Overall, our combined results suggest that the SOM in the Sargasso Sea arises as a recurrent biogeochemical feature from the combination of seasonal shifts in physical water column structure and enhanced microbial activity, at least partly mediated by viral infection. Critically, our analysis of -omics, polony/iPolony, and historical data demonstrate how the viral shunt may play a central role in driving SOM biogeochemical dynamics. As stratification is expected to intensify and deep mixing anticipated to weaken due to increased annual sea surface temperatures[58,59], we expect emergent effects on the formation and relative importance of the SOM, and consequently the viral shunt, to biogeochemical fluxes of carbon and oxygen[60]. Our results emphasize the need to incorporate viral infection into changing ocean modeling and prediction as a critical mechanism for understanding carbon remineralization dynamics in the subsurface of oligotrophic oceans. We recognize that our inference focuses on a particular study site in the oligotrophic Sargasso Sea and that insights derived from high-resolution temporal sampling in archetypal ocean basins remain limited. Precisely so, we suggest that future efforts should include targeted quantification of bulk- and single-cell viral infection, using a combination of high-throughput (e.g., single-cell transcriptomics) and host-resolved laboratory infection assays (e.g., iPolony) to more accurately quantify viral contributions to SOM biogeochemistry. Such efforts will enable better modeling and prediction of mutual feedbacks between climate variability, SOM formation, and the macroscale function of SOMs.

## Methods

### Sampling design and AE1926 CTD data processing
At the onset of the study, a surface buoy with an underwater drogue (at ~30 m depth) was deployed to allow us to follow the same "patch" of water for the duration of the study. Water samples were collected using a CTD-rosette equipped with 24 ×12-L Niskin bottles. Depths for each cast were chosen based on in situ CTD oceanographic parameters to allow for focus on water column features: this included the top (surface, surface) and bottom (base of the mixed layer) of the upper mixed layer, one depth within the highly oxygenated zone below the mixed layer (subsurface oxygen maxima, SOM), and one depth within the zone with the highest chlorophyll fluorescence values (the deep chlorophyll maxima, DCM). For each cast the CTD was deployed to at least 500 m to collect data on physical water column structure. CTD profiles used in Fig. 1 are based on the water column profiles collected every 4 h during the sampling campaign. Water samples were collected from four main depths every 12 h starting at 8:00 ADT (GMT-3): ~5 m depth (surface), ~40–50 m depth (base of the mixed layer), ~42–60 m depth (SOM) and 105–120 m depth (DCM).

CTD data were retrieved as deposited in BCO-DMO[61]. Measurements were binned to the nearest 0.5 m of depth to standardize across casts, then a Nadaray-Watson kernel smoothing filter with bandwidth of 5 db was applied to each variable to remove noise and spikiness. Mixed layer depth for each cast was calculated using the criterion of an increase in potential density of 0.125 from a reference pressure of 10db to account for instrumental noise in the surface[62–65]. Notably, this mixed layer depth criterion tends to identify the potential density surface directly below which oxygen saturation % increases to over 100%, operationally capturing the SOM depth as the first isopycnal (within $0.1\,kg/m^3$) below the MLD as defined using the $0.125\,kg/m^3$ change from surface criterion. This mixed layer depth criterion is presented as a preset commonly used in oceanographic timeseries data archives[23]. For statistical comparison of depth layers and depth layer-integrated time series analysis, smoothed data were averaged into depth bins representative of those layers – the mixed layer was binned from 5 to 40 m, the SOM from 60 to 50 m, and the DCM from 120 to 100 m. Beam attenuation, chlorophyll fluorescence, oxygen concentration, and oxygen saturation were all averaged (mean) across these layers. The beam attenuation measures particles within the 0.5–20 µm size range[21,22]. For diel periodicity analysis, data were initially detrended using a linear model, and then the nonparametric rhythmicity detection method 'rain' v.1.30.0[66] was run in 'independent' mode with a test period of 24 h.

### BATS CTD data processing
CTD cast data from October 1988 through December 2019 were downloaded from the API available at http://bats.bios.edu/bats-data/ in ASCII format. Bottle data were also downloaded from the API available at batsftp.bios.edu/BATS/bottle/bats_bottle.txt in ASCII format. Data on experimental rate measurements from BATS isotope incorporation experiments were also downloaded from the BATS API (currently available via the dropbox link: https://www.dropbox.com/scl/fo/x7xvlmyzqh9t9tfpe8ffg/AEBrkFFxVFvlO31MJmd9pcY?rlkey=7v73mfwhd78fdtxvfhk4v4kca&e=1&dl=0). Rate data were matched with the corresponding CTD cast in the BATS CTD data record and experiments taken between May and October (the months where stratification typically occurs) conducted above 200 m were used for further analysis. After initial reformatting to account for inconsistencies in file formats between casts, any cast with faulty or missing data for conductivity, temperature, or pressure, as well as casts missing data from the top 10 db of pressure were removed so that all casts could have the same thermodynamic calculation processing. The remaining 5270 CTD profiles were then processed using methods from the Gibbs Seawater Toolbox (https://www.teos-10.org/pubs/gsw/html/gsw_front_page.html) using the python 3.4.0 distribution through R via the reticulate package v1.24 (https://github.com/rstudio/reticulate). Briefly, height from surface of geoid, potential salinity, absolute salinity, conservative temperature, potential density (sigma-theta), oxygen solubility, and oxygen saturation were calculated using pressure, longitude, latitude, temperature, conductivity, and oxygen CTD data.

Mixed layer depth was calculated as described above. Bottle data, including flow cytometry data, were also downloaded from the BATS API and matched to corresponding casts and depths. The quality of CTD optode data was assessed through a systematic comparison of CTD oxygen values to bottle measurements using the Winkler method[67]. A linear regression was used to assess the correspondence between methods. The fit was $[O_2 \ \mu M$ via CTD optode] $= 7.85 + 0.964*[O_2 \ \mu M$ via Winkler method] with an adjusted $R^2 = 0.959$. Any cast with a measurement containing a standardized residual of 3 or greater, indicating a large discrepancy between CTD optode and bottle oxygen measurements, was discarded for further analysis (total of 26/5270 casts).

## Prochlorococcus-virus seasonal relationship modeling

VLP counts associated with BATS monthly time series cruises from Parsons et al., were retrieved from supplementary information and matched to corresponding BATS time series cruise bottle and CTD data, processed as described in the above section. Altogether, 104 observations were paired resulting in Prochlorococcus flow cytometric counts from BATS publicly available bottle data, oxygen concentrations from BATS CTD data, and VLP counts. Data were separated by month to estimate monthly varying effects on the relationship between Prochlorococcus and VLPs. For each month, a type-II regression model (major axis estimation procedure) was fit using the R v4.2.1 package lmodel2 v.1.7-3[68] for $\log_{10}$ Prochlorococcus counts and $\log_{10}$ VLPs. The slope of this linear model on log-log data is equivalent to the exponent of a power-law relationship between Prochlorococcus counts and VLP counts. Parameter estimate uncertainty and model significance were assessed using a permutation test with 10,000 permutations per model.

## Metatranscriptome sampling, processing, and bioinformatic analysis

Seawater was filtered through a 0.2-μm pore-size Sterivex™ filtration unit. Residual water was removed by pushing air through the filter with a 60 ml syringe, and the filter immediately transferred to a −80 °C freezer within 1 h of sample collection. RNA was extracted using a publicly available phenol-chloroform based protocol[69] with DNA contamination removed using the Turbo DNA-free™ kit (Ambion®). Metatranscriptome libraries were prepared by reducing ribosomal RNA using the QIAGEN's FastSelect kit (5S/16S/23S for bacterial rRNA depletion) and sequenced (2 × 151 nt) using the low-input protocol for total RNA on the Illumina Novaseq S4 platform under the DOE Joint Genome Institute (JGI) Community Sequence Proposal ID# 505733. For samples with <10 ng of RNA total, the ultra-low input protocol was used (13 PCR cycles versus 10 cycles for standard low-input). Specific library preparation methods used for each sample can be found on the JGI Genome Portal (Project ID 505733).

Raw read filtering and trimming were done using BBDuk v38.67 and BBMap v38.84 from the BBtools packages[70]. Trimmed filtered reads were combined across samples and assembled using MEGAHIT v1.2.9[71]. MetaGeneMark v3.38 was used to call open reading frames (ORFs) with a kmer size parameter as --k-list 23,43,63,83,103,123. For the cellular community, trimmed filtered reads were mapped to the combined assembly using BBMap v38.84[70] with default parameters, and tabulated using featureCounts from the Subread package v. 2.0.1[72]. ORF protein sequences were annotated using eggNOG-mapper v2.1.4[73] for functional annotation, and aligned to the PhyloDB database (https://github.com/allenlab/PhyloDB) using the software package EUKulele v.2.1.0[74] for taxonomic annotation of bacteria, eukaryotes, and archaea. ORFs were filtered with average read counts ≥10 across the dataset. Variance stabilizing transformation (VST) using the DESeq2 v.1.34.0 R-package was used for read normalization[75].

We used the Kyoto Encyclopedia of Genes and Genomes (KEGG) orthology annotations[31] to explore actively transcribed metabolic pathways. Normalized read counts for each KEGG orthologue were summed for the genes assigned to each taxon within each sample, resulting in order-level signals for heterotrophic bacteria, genus-level for cyanobacteria, and class-level for eukaryotes. To assess differences between depth/time, we calculated the ratio of each KEGG KO VST to the corresponding order-level summed VST of rpoB/RPB1's (DNA-directed RNA polymerase subunit-beta/large subunit, K04043 and K03006) which assesses relative taxonomic contribution to each sample[76], resulting in a "KO vst:rpoB ratio" to correct for positive correlations between increases in gene abundance with taxon abundance. This approach has been recommended for adjusting the gene transcript abundance to the taxon-level RNA estimate within each sample[77] when paired metagenomic data are not available.

## Metatranscriptome recruitment to Single-Celled-Amplified genome (SAGS) of Prochlorococcus

To resolve spatial and temporal patterns in Prochlorococcus ecotypes, we competitively recruited metatranscriptome reads to publicly available SAGS from Berube et al.[78]. SAG genomic assemblies, gene annotations from the IMG Annotation Pipeline version 4 (accommodating phylogenetic information pertaining to ecotype and clade of each SAG) and the cyanobacterial concatenated phylogenetic tree were retrieved via:

https://figshare.com/articles/dataset/File_12_Genome_sequences_and_annotations/6007223?backTo = /collections/Single_cell_genomes_of_i_Prochlorococcus_i_i_Synechococcus_i_and_sympatric_microbes_from_diverse_marine_environments.

Trimmed, filtered reads were competitively mapped to a concatenated file containing all SAG genomic assemblies using BBMap v38.84[70] with default parameters and tabulated using the GFF file gene coordinates with featureCounts[72]. Only SAGS that had ≥10% of their genes mapped to transcripts were considered 'detected' to avoid capturing spurious read recruitments[79] and genes with <20 reads mapped across the entire time series were filtered out. Reads were normalized using the VST method[75].

## Statistical analysis of metatranscriptome data

Statistical analyses were performed using the R Statistical Software v4.1.3[80]. Principal components analysis (prcomp) in R) was performed on VST normalized values across all ORFs detected in the metatranscriptome assembly. The Kruskal-Wallis test (kruskal.test via stats v.4.3.1) followed by Dunn's multiple comparison (dunn.test v1.3.5) was used to determine significance of trends between depths (surface, base of the mixed layer, SOM, DCM), separating 08:00 and 20:00 time points. The Benjamini−Hochberg adaptive false discovery rate (FDR) control procedure was implemented using a significance threshold of FDR = 10% ($P < 0.1$). To be considered significantly elevated or depleted at the SOM, the mean VST value (KO VST:rpoB ratio) at the SOM had to be either higher or lower (with an adjusted p-value of ≤ 0.1) than the surface, base of the mixed layer and DCM. Fold-change values for vOTU scaffolds harboring DNA_Pol_A were done using DESeq2 v.1.34.0[75] on scaffold-summed raw counts for the following comparisons: SOM versus surface and SOM versus DCM samples collected at 8 AM.

## Virome sample collection, processing, and bioinformatic analysis

Seawater (10 L) was 0.22-μm filtered to remove bacteria, and the remaining viruses then concentrated from the filtrate using iron chloride flocculation[81] followed by storage at 4 °C. Filters were cut in half and viruses resuspended from one half in ascorbic-EDTA buffer (0.1 M EDTA, 0.2 M Mg, 0.2 M ascorbic acid, pH 6.0). Viral particles were concentrated using Amicon Ultra 100 kDa centrifugal devices (Millipore), treated with DNase I (100 U/mL) followed by the addition of 0.1 M EDTA and 0.1 M EGTA to halt enzyme activity[20], and DNA was

extracted using Wizard PCR Preps DNA Purification Resin and Mini-columns (Promega, Cat. #A7181 and A7211 respectively) after Henn et al.[82].

All samples from the BATS virome were sequenced at ~144 M reads per sample on an Illumina NovaSeq S4 platform at the DOE Joint Genome Institute. Raw reads from all 39 samples in the BATS viromes dataset went through quality control using BBDuk (https://jgi.doe.gov/data-and-tools/software-tools/bbtools/). Adaptors and Phix174 reads were removed (ktrim=r minlength = 30 $k$ = 23 mink = 11 hdist = 1 hist2 = 1) and reads trimmed (qtrim=rl maq=20 maxns=0 minlength=30 trimq=20). Reads were assembled individually using MegaHIT 1.2.9[71] and those ≥1.5 kbp in length were piped through VirSorter2 SOP version 2.2.3[83] and CheckV v0.8.1[84] following the viral sequence identification SOP[85]. Resulting viral contigs were clustered into viral populations (vOTUs) at ≥95% identity and ≥80% coverage using ClusterGenomes (https://github.com/simroux/ClusterGenomes). This resulted in a total of 44,819 vOTUs of which 13,369 were ≥10 kbp. After dereplication, contigs were piped through Virsorter2 for the second time with the --prep-for-dramv flag. The contigs were annotated using the DRAM v.1.4.0 annotator (min-contig-size = 1000, --skip_trnascan) for viromes (https://github.com/WrightonLabCSU/DRAM). Trimmed filtered metatranscriptome reads were mapped to the ≥10 kbp vOTU database using BBMap v38.84[70] with the minimum read identity set to 95% and normalized using the VST method[75].

To assess broad phylogenetic associations of vOTU contigs, T4-like and T7-like phage hallmark proteins were searched across the ≥5 kbp vOTU annotations. The following PFAMs were used for identifying T4-like contigs: gp23 (T4 major capsid protein, PF07068), gp32 (DNA-binding protein, PF08804), gp45-slide_C (sliding DNA clamp, PF09116), GPW_gp25 (tail sheath gpW/gp25-like domain, PF04965), Phage_gp53 (Base plate wedge protein 53, PF11246), Phage_sheath_1 (Phage tail sheath protein subtilisin-like domain, PF04984), Phage_T4_gp19 (T4-like virus tail tube protein gp19, PF06841), T4_baseplate (T4 bacteriophage base plate protein, PF12322), T4_gp59_N (gp41 DNA helicase, PF08993), T4_gp9_10 (Baseplate wedge protein gp10, PF07880), T4_neck-protein (Neck protein gp14, PF11649), T4_tail_cap (tail-tube assembly protein gp48, PF11091), T4-gp15_tss (T4-like virus Myoviridae tail sheath stabilizer, PF16724). The following PFAMs were used for identifying T7-like contigs: Phage_T7_Capsid (Phage T7 capsid assembly protein, PF05396), Phage_T7_tail (Phage T7 tail fibre protein, PF03906).

Phylogenetic trees of DNA polymerase A (DNA_pol_A, PF00476) and T4 Major capsid proteins (gp23, PF07068) predicted from the vOTU contigs were constructed, with references downloaded from NCBI RefSeq representative of diverse phage families. Here, vOTU DNA_pol_A sequences >700 aa or gp23 sequences >450 aa in length were aligned to reference sequences in MEGA7[86] using ClustalW[87] and trimmed using trimAl (v.1.2; -gappyout method)[88]. The maximum-likelihood tree was constructed in PhyML 3.0[89] with the LG model, and the Shimodaira-Hasegawa (SH)-like approximate likelihood ratio test (aLRT-SH-like). Remaining DNA_pol_A sequences <700 aa or gp23 sequences <450 aa were placed on the tree using pplacer[90] and visualized and annotated using ITOL v.4[91].

### Flow cytometry and statistical analysis

Samples for flow cytometry were collected in duplicate after pre-filtration through a 20 μm mesh and fixed in 0.125% glutaraldehyde for 15 min in the dark, frozen in liquid nitrogen, and stored at −80 °C until analysis. The samples were run using an Influx DB flow cytometer (BD Biosciences) equipped with a small particle detector, a 488-nm and a 457-nm laser, and a 70-μm nozzle tip. Samples were weighed to determine the volume analyzed by the sorter. *Synechococcus* and *Prochlorococcus* cell abundances were determined based on autofluorescence and size. *Prochlorococcus* was detected by red fluorescence of chlorophyll $a$ (emission at 692/640 nm) while *Synechococcus*

was detected by orange fluorescence of phycoerythrin (emission at 580/30 nm, Supplementary Fig. 12). Total bacteria were enumerated by staining with $10^{-4}$ diluted stock of SYBR Green I (Invitrogen), followed by a 15 min dark incubation. Total bacteria were detected using green fluorescence excited with the 488 nm laser and detection at emission wavelengths of 530/20 nm. Heterotrophic bacterial counts were quantified as the difference between the total bacteria less the abundance of the cyanobacteria. To determine infection levels of *Prochlorococcus* by the iPolony method (see below), *Prochlorococcus* were sorted using 1.0 drop purity mode with the same parameters described above (Mruwat et al.). The BD FACSDiva v8 software was used to process the flow cytometery data. Using Prism (v 9.5.1), the Kruskal-Wallis test followed by Dunn's multiple comparison was used to determine significance of trends across depth (surface, base of the mixed layer, SOM, DCM) for each taxonomic group, integrating time. The Benjamini–Hochberg adaptive false discovery rate (FDR) control procedure was implemented using a significance threshold of FDR = 10% ($P < 0.1$).

### Polony detection of cyanophage and statistical analysis

Samples for cyanophage enumeration were filtered through a 0.2 μm syringe filter and the filtrate was frozen at −80 °C without fixative. Quantification of T4- and T7-like cyanophage abundances was done using the polony method following Baran et al.[18]. for T7-likes and Goldin et al.[92]. for T4-likes. Degenerate 5′-acrydite-modified primers and the 0.2 μm filtrate from seawater samples were incorporated into an acrylamide gel prior to polymerization within a 40 μm-deep well etched into microscope slides (Thermo Fisher Scientific). Subsequently, other PCR reagents and the unmodified degenerate primer were diffused into the gel post-polymerization. DNA amplification was conducted using slide PCR, and amplicons were detected through hybridization with fluorescent probes using a GenePix 4000B microarray scanner (Axon Instruments). The primers and probe for T7-like cyanophages targeted the DNA polymerase gene (*DNApol*), for both clade A and clade B phages and are reported together as T7-like cyanophages. Notably, clade A cyanophages were found to be a minor contributor to T7-like cyanophages in this study, and they are reported together with clade B. Cyanophages belonging to the newly described clade C were not quantified using this method since they lack the DNA polymerase gene[93]. For T4-like cyanophage quantification, the primers and probes targeted the *g20* portal protein gene.

Samples for quantifying infection were collected and fixed as described for flow cytometry analysis. To evaluate the direct impact of viruses on cyanobacteria, we employed the iPolony method[19] which analyzes the presence of viral DNA within cyanobacteria cells. In this approach *Prochlorococcus* cells are sorted by flow cytometry (see above) and immediately embedded in polyacrylamide gels at an average of four thousand cells per gel. Subsequently, virus DNA was amplified and detected within the cells using the same amplification and hybridization procedures as those described for free cyanophages, with the exception that cells were not pretreated with EDTA for analysis of infection by T4-like cyanophages. For further details refer to Mruwat et al.

Using Prism (v 9.5.1), the Kruskal-Wallis test followed by Dunn's multiple comparison was used to determine significance of trends across depth (surface, base of the mixed layer, SOM, DCM) for T7- and T4-like polony or iPolony values, integrating time. The Benjamini–Hochberg adaptive false discovery rate (FDR) control procedure was implemented using a significance threshold of FDR = 10% ($P < 0.1$).

### Reporting summary

Further information on research design is available in the Nature Portfolio Reporting Summary linked to this article.

## Data availability

Raw metatranscriptomic and viromic sequencing data is available through the JGI under project ID #505733 and deposited into the NCBI SRA IDs SRR34589202- SRR34590687. The full accession numbers and quality read counts are also provided in Supplementary Data 5. All cruise data is available via BC-DMO (See Wilhelm et al.[61]). Source data for each figure are provided with this paper.

## Code availability

Code used for analyses and generation of figures presented in this paper are available on github at the url https://github.com/d-muratore/virus_o2max. A release of this repository has an available DOI at https://doi.org/10.5281/zenodo.17652303[94]. All intermediate data files and public datasets used to generate manuscript figures and present results are available for separate download to be incorporated into the repository to reproduce code on figshare, the figshare data are all permanently archived and available at the DOI:10.6084/m9.figshare.30047524 (https://doi.org/10.6084/m9.figshare.30047524.v1).

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

## Acknowledgements

The authors would like to thank the captain and crew of the RV *Atlantic Explorer* for conducting cruise AE1926, Rod Johnson and the entire BATS team for advice and assistance, Yotam Hulata for helping with flow cytometry, Shay Kirzner for helping with *Prochlorococcus* sorting, and Miranda Harmon-Smith, Emily Eloe-Fadrosh and Simon Roux for help with the JGI CSP project. This work was funded by NSF grant OCE-1829641 to S.W.W., Simons Foundation grant 735077 to S.W.W., NSF grant OCE-1829636 to J.S.W., Simons Foundation grant 721231 to J.S.W., the Blaise Pascal Institute Chair of Excellence award at the Institut de Biologie of the École Normale Supérieure to J.S.W, Simons Foundation Life grants 529554 and 735081 to D.L., Israel Science Foundation grant 2679/20 to D.L., NSF grant OCE-1737237 to A.B., and the Omidyar Complexity Postdoctoral Fellowship (by the Santa Fe Institute) to D.M. Sequencing was provided by the Joint Genome Institute CSP grant 505733 to M.S. and C.S. This work was performed under the auspices of the U.S. Department of Energy by Lawrence Livermore National Laboratory under Contract DE-AC52-07NA27344.

## Author contributions

S.W.W. and J.S.W designed the study. N.E.G., D.M., C.S.G, A.C., K.S.N., and C.L.S. analyzed data. N.E.G., D.M., G.R.L., C.S.G, S.M.C, H.L.P., A.C., I.M., J.M.D., A.B., A.R.C., L.C. and S.W.W. collected and processed field samples. N.E.G. and D.M. wrote the manuscript. G.R.L., D.L., M.B.S., J.S.W. and S.W.W edited the manuscript.

## Competing interests

The authors declare no competing interests.

## Additional information

¹Department of Microbiology, The University of Tennessee, Knoxville, TN, USA. ²School of Biological Sciences, Georgia Institute of Technology, Atlanta, GA, USA. ³Faculty of Biology, Technion-Israel Institute of Technology, Haifa, Israel. ⁴Department of Microbiology, The Ohio State University, Columbus, OH, USA. ⁵School of Physics, Georgia Institute of Technology, Atlanta, GA, USA. ⁶Biosystems Engineering and Soil Science, The University of Tennessee, Knoxville, TN, USA. ⁷Department of Civil, Environmental, and Geodetic Engineering, The Ohio State University, Columbus, OH, USA. ⁸Department of Biology, University of Maryland, College Park, MD, USA. ⁹Department of Physics, University of Maryland, College Park, MD, USA. ¹⁰University of Maryland Institute for Health Computing, North Bethesda, MD, USA. ¹¹Present address: Physical and Life Sciences Directorate, Lawrence Livermore National Laboratory, Livermore, CA, USA. ¹²Present address: Santa Fe Institute, 1399 Hyde Park Rd, Santa Fe, NM, USA. ¹³These authors contributed equally: Naomi E. Gilbert, Daniel Muratore. ✉e-mail: jsweitz@umd.edu; wilhelm@utk.edu

