## [Transparent Peer Review file · Nature Communications]

Seasonal Enhancement of the Viral Shunt Catalyzes a Subsurface Oxygen Maximum in the Sargasso Sea

Corresponding Author: Professor Steven Wilhelm

Version 0:

Reviewer comments:

Reviewer #1

(Remarks to the Author)

In "Seasonal Enhancement of the Viral Shunt Catalyzes a Subsurface Oxygen Maximum in the Sargasso Sea," Gilbert et al used transcripts, polonies for cyanophage abundance, and ipolonies for picocyanobacterial infection on a cruise in the Sargasso Sea in October 2019. They combine this data with a re-analysis of BATS oxygen and viral-like particle timeseries data. The purpose of the paper is to link Subsurface Oxygen Maxima (SOM) to Prochlorococcus productivity and viral lysis. They found increased numbers of Prochlorococcus cells, cyanophage viruses, infected Prochlorococcus cells and cyanophage transcriptional activity at the SOM. In the literature, viral lysis is often linked to biogeochemistry in a vague hand-wavy way, so any paper that can link the two in a concrete fashion is very cool. I was happy with the bioinformatic analyses and the polonies are always interesting and useful. This paper is very interesting, but it could be pushed a bit farther to be a better paper. Particularly, the authors should incorporate existing primary productivity and bacterial productivity measurements from BATS to see if that data upholds their hypothesis. Also, rather than simply dismissing the physics aspects, it would be good if the physics was included in a small way (see below). And please make sure that the DOE/JGI sequencing is submitted to NCBI and acknowledged appropriately.

One thing that I found confusing about this paper was that it starts sounding like it is known that the Sargasso Sea and BATS have SOMs, but then the first thing the authors do is prove that BATS has SOMs. Chen and McKinley (2019) analyzed 26 years of BATS data plus floats. It would good to know what this current paper did that was different. I think they just had to do the work over again because the authors are non-overlapping.

On that same note, the authors have set up a physics versus biology argument in the Introduction, and they are clearly on the biology side. However, the real ocean isn't physics or biology, it is physics and biology. Chen and McKinley (2019) say that "After the spring bloom, surface waters that are enriched in oxygen and organic matter, but low in nitrate, are subducted and transported along the seasonal isopycnals that progressively displace downward." So physics is definitely happening. And the authors have the timeseries data to see that the physics is happening. It would be great if this could be a physics AND biology type of paper. How much more oxygen was there in August compared to when the water subducted in the spring?

Also, the question is really about primary production rates versus rates of heterotrophy. While it does not seem that these rates were measured on the cruise in question, primary production rates are measured at BATS. Are the rates higher or lower in the SOM below the mixed layer compared to mixed layer? Bacterial production is also measured at BATS, or at least was in Steinberg et al 2001. Please incorporate these data into your arguments. Are rates of primary production greater than bacterial production? This information is critical to understanding what is going on in the SOM.

Detailed comments

I received this paper to review from Nature Communications. My impression is that Nature Communications has a requirement for separate Results and Discussion sections (though I defer to the editor on this). This paper needs Results and Discussion headings.

Lines 56-61: ipolonies, numbers of infected Prochlorococcus, were left out of this sentence. However, they are the smoking gun!!! Please add in.

line 87: the references are for the phototrophic populations and there are no citations for the oxygen concentrations. I guess Chen and McKinley 2019 should be cited here.

Line 100: Could this be elaborated on? What exactly is still elusive? The paragraph kind of makes it sound solved already. There is some room here to elaborate.

Line 102: use a comma instead of a dash

Line 155: List the range of years that these CTD casts come from: (1988-2019). Readers get to the methods last in these papers and so need some help.

Lines 171-172: You could quantify this. How much more oxygen was there in August compared to when the water subsided?

Line 179: Order by depth: upper mixed layer, SOM, DCM

Line 183: The abbreviations are getting excessive here. I think you can write out the word surface. Also, just write out base of the mixed layer. Excessive abbreviation are a barrier to reader understanding.

Line 197 etc: using metagenomic data, (Fuchsman and Hays, 2023; Fuchsman et al., 2023) clearly show the same *Prochlorococcus* ecotypes in the mixed layer and right below the mixed, but different ecotypes dominating the DCM in the subtropical North Atlantic. Also some good information about cyanophage dynamics.

Line 210: were. Past tense

Line 219: I am not sure what sub-surface mixed layer peaks means. I mean, I figured it out, but maybe explain more clearly.

Line 228-230: Really you want primary production rates and bacterial production rates for these analyses. Not cell numbers.

Line 254 and lines 268-274: There is usually an ammonium peak at the DCM or right below the peak of the DCM. So this is easily explained. You can see such an ammonium peak in (Widner et al., 2018) in the Pacific, but there must be similar data from the Atlantic. I believe it is a typical feature in the ocean.

Woodward and Rees (2001) Deep Sea Research Part II shows the ammonium peak in the Atlantic. There is also Rees et al 2006 Deep Sea Research Part II.

I think you can assume this feature is also present at BATS.

Line 259: lower than the mixed layer? Or the DCM?

Line 274: That is really too bad!

Line 311: Also seen in (Fuchsman and Hays, 2023)

Line 344: viral isolates

Line 432: lol! I am sure you can find a paper than uses these criteria, which would be better than a webpage. Probably one of Dave Karl's tomes. Anyway, fix this.

Karl et al 2021 says "depths of upper mixed layer based on the density criterion of de Boyer Montegut et al. (2004)," so maybe track down that one.

Line 435: List the shallower depths first

Line 454-455: You already said how you calculated the mixed layer above.

Line 478-480: Do you have a time estimate of how long it took to get the RNA from the Niskin bottles to the freezer? This is critical for determining quality.

Line 644: This sequence data needs to be submitted to NCBI SRA. The people at JGI will do this for you. But it needs to be done. And you need to add a supplemental table with all of the accession numbers and numbers of good quality reads.

Line 644: The BATS bottle data should also be specifically listed out here.

Line 658: I also have a DOE CSP grant and was told to also include the award doi in publication. I think you should also be acknowledging the staff that helped with sequencing and sequence analysis. Please look at the directions related to your award.

Please put the supplemental figures and their figure captions into one combined word document!!!!!!!!!!!!!!!!!!!!

Since this is now a Nature Communications Document, the authors could take a few of the many supplemental figures and make them main document figures. I would recommend moving Figure S2 (SOM data from BATS) and Figure S4 *Prochlorococcus* ecotypes. It clearly shows that the SOM is more similar to the mixed layer than to the DCM.

Bibliography

Fuchsman, C.A., Garcia Prieto, D., Hays, M.D., and Cram, J.A. (2023) Associations between picocyanobacterial ecotypes and cyanophage host genes across ocean basins and depth. *PeerJ* 11: e14924.

Fuchsman, C.A. and Hays, M.D. (2023) Increased cyanophage infection at the bottom of the euphotic zone, especially in the fall. *Environ Microbiol* 25: 3349–3363.

Widner, B., Mordy, C.W., and Mulholland, M.R. (2018) Cyanate distribution and uptake above and within the Eastern Tropical South Pacific oxygen deficient zone. *Limnol Oceanogr* 63: S177–S192.

Reviewer #2

(Remarks to the Author)

Gilbert et al. is a study examining the role of viruses in the formation of SOMs in water at the BATS site. Using geochemistry, transcriptional activity, and cell counts over timeseries they conclude that viral predation enhances oxygen production and NPP. The manuscript is well written, and the approaches are adequately support the conclusions. There are no major issues with this study.

Specific issues:

Ln 85 - Suggested re-wording to remove the repetition of "by" in this sentence.

Ln 260 - Suggested change to: that were significantly

Ln 272 - Are the strains in the SOM part of this group with constitutively high amt transcription?

Ln 287 - Are there references that could be cited here to back up this assumption that transporters are induced in the presence of their substrates?

Ln 432 – missing citation

Ln 435 - This doesn't matter for understanding but this sentence may read more easily if all the layers are listed from either minimum to maximum (i.e. 5-40 m, 100-120 m, 50-60 m) or max-min.

Ln 551 - Which Illumina instrument was used?

Ln 558 - The recommendation for clustering on large-scales (>1,000s of sequences like you have here), according to the ClusterGenomes github page, is to use anicalc and aniclust in the CheckV distribution instead of ClusterGenomes.

I would recommend to re-run anicalc and aniclust from CheckV and verify the ClusterGenomes clustering accurately clustered the large number of viral sequences from this study.

Reviewer #3

(Remarks to the Author)

Version 1:

Reviewer comments:

Reviewer #1

(Remarks to the Author)

This paper is much improved. As I said previously, this work is noteworthy for actually linking viral infection and biogeochemistry. The results are excellent. I list out small problems below.

Details

Line 68 Abstract: hmmm.... Usually people say that increased transcription of transporters indicates limitation. Here you say the opposite. I might edit or cut this sentence.

--The main text deals with this issue very well. Perhaps, in the abstract, expand the sentence to say elevated amt combined with lower ntcA transcripts indicate a response to the presence of increased ammonium.

Line 98: Change comma to period.

Line 151: "The SOM during this study" – at first I thought you were referring to the SOM in the paper described in the previous sentence. Perhaps "our study"

Line 169: You work in scientific papers should be in the past tense. Identified

Line 173: Indicated

Line 188: correlated

Line 190: measure is used awkwardly here

Line 227-228: The sentence "A peak in Prochlorococcus HLI has been observed below the mixed layer in the Mediterranean Sea." Did that occur during a SOM? Yes, it did. See Haro-Moreno et al 2018 "Fine metagenomic profile of the Mediterranean..." Table 1.

Line 277-278: Diel cycling is reduced at the DCM in general. See Vislova et al 2019. "Diel oscillation of microbial gene transcripts declines with depth in oligotrophic ocean waters"

Lines 311-313: This is nice.

Lines 394-396: And primary production!!!! Add in.

Line 411: but Prochlorococcus always increases amt expression at dusk??? This is normal. Vislova et al 2019.

Line 682: Raw meta-transcriptome and virome reads should be deposited at NCBI SRA. This is a community standard.

It is possible that Table 1 should be supplemental just do to the fact that it is so big.

Figure 1c—red green color blind people can't read this one.

Fig 4bc—same

I know that red green color-blind people seem theoretical, but I have two in my lab. It is more common than you think.

Reviewer #2

(Remarks to the Author)

I feel all my comments were addressed adequately.

Reviewer #3

(Remarks to the Author)

REVIEWER COMMENTS

Reviewer #1 (Remarks to the Author):

In “Seasonal Enhancement of the Viral Shunt Catalyzes a Subsurface Oxygen Maximum in the Sargasso Sea,” Gilbert et al used transcripts, polonies for cyanophage abundance, and ipolonies for picocyanobacterial infection on a cruise in the Sargasso Sea in October 2019. They combine this data with a re-analysis of BATS oxygen and viral-like particle timeseries data. The purpose of the paper is to link Subsurface Oxygen Maxima (SOM) to Prochlorococcus productivity and viral lysis. They found increased numbers of Prochlorococcus cells, cyanophage viruses, infected Prochlorococcus cells and cyanophage transcriptional activity at the SOM. In the literature, viral lysis is often linked to biogeochemistry in a vague hand-wavy way, so any paper that can link the two in a concrete fashion is very cool. I was happy with the bioinformatic analyses and the polonies are always interesting and useful. This paper is very interesting, but it could be pushed a bit farther to be a better paper. Particularly, the authors should incorporate existing primary productivity and bacterial productivity measurements from BATS to see if that data upholds their hypothesis. Also, rather than simply dismissing the physics aspects, it would be good if the physics was included in a small way (see below). And please make sure that the DOE/JGI sequencing is submitted to NCBI and acknowledged appropriately.

We thank the reviewer for the comments and feedback. We have endeavored to address all their critiques in the revision (and address them point by point below).

One thing that I found confusing about this paper was that it starts sounding like it is known that the Sargasso Sea and BATS have SOMs, but then the first thing the authors do is prove that BATS has SOMs. Chen and McKinley (2019) analyzed 26 years of BATS data plus floats. It would be good to know what this current paper did that was different. I think they just had to do the work over again because the authors are non-overlapping.

We agree: the presence of an SOM in the Sargasso has been known for some time as the author points out - both Steinberg et al (2001) and Parsons et al (2011) (both cited in the paper). Like Chen and McKinley (now cited in the revision), these authors display the data across the calendar year, averaged by month. We reanalyzed oceanographic climatological data collected from 1988 to 2019 to contextualize our October 2019 cruise data within the physical and biogeochemical context of the BATS SOM. In the present case we also provide the data by month in order to show the consistency of the result (see Figure 2B). Notably, while Chen and McKinley use 26 years of BATS bottle

oxygen data. In our analysis, we focus on the 10m below the mixed layer depth and expand the record for analysis by using a standardized method of sensor data processing and thermodynamic conversions, calibrated against bottle oxygen measurements using the Winkler method (see Methods) – resulting in 5,072 CTD casts with high-quality oxygen profiles spanning 31 years. This gives our analysis higher depth resolution over the full time series (see Figure 2B, Figure 3). We have now elaborated on these points, clarified the context, and provided more information on the methodological difference that we then leverage to contextualize the October 2019 cruise (Lines 157-184).

On that same note, the authors have set up a physics versus biology argument in the Introduction, and they are clearly on the biology side. However, the real ocean isn't physics or biology, it is physics and biology. Chen and McKinley (2019) say that "After the spring bloom, surface waters that are enriched in oxygen and organic matter, but low in nitrate, are subducted and transported along the seasonal isopycnals that progressively displace downward." So physics is happening. And the authors have the timeseries data to see that the physics is happening. It would be great if this could be a physics AND biology type of paper. How much more oxygen was there in August compared to when the water subducted in the spring?

We agree with the reviewer that the SOM has both physical and biological components. The central goal of the present paper is to elucidate mechanisms underlying the biological component of the biogeochemical change, which the reviewer supportively mentions we do 'in a concrete fashion'. Our study reveals mechanistic evidence linking the viral shunt mechanism to SOMs. This does not preclude an analysis of the physics in future work, but as we have argued, connecting an enhanced viral shunt to SOM persistence is both novel and ecologically relevant.

That said, from our reanalysis of an updated version of the Chen and McKinley review of the BATS history (now with an additional 5 years of data included in this paper), we found two peaks in isopycnal oxygen content and oxygen saturation from isopycnals that lie between the average mixed layer density in the stratified period (< about 1024.4 kg/m³ from our analysis) and those that subduct below the SOM depth (about 1024.6 kg/m³) - the first peak we associate with initial ventilation upon deep mixing (March peak) followed by a second peak post-subduction in July that is on average 2-5% higher O₂ concentration immediately post-ventilation. We argued that this secondary accumulation in oxygen total concentration and saturation likely has a biological component, given that the increase occurs following the annual timing of restratification. Chen and McKinley, who strictly invoke a physical mechanism of isopycnal subduction to explain seasonal dynamics in subphotic oxygen at BATS, indeed find evidence

consistent with our hypothesis that enhanced oxygen production occurs only in an extremely specific physical environment directly beneath the mixed layer depth, where something different is occurring than the net heterotrophy they propose for the subducted isopycnals at slightly deeper depths. These data are presented in their main text figure 3e, where it is clear that the along isopycnal change in O_2 anom, in the small (5-10m) window directly beneath the mixed layer depth during the stratified period (the red shaded area directly underneath the MLD marked with the black line), there is a positive change, unlike the negative change underneath it associated with the broader 50-100m depth strata that they focus on for their broader analysis.

We have reworked the paragraphs in lines 157-197 to reflect our analysis that sustained O_2 saturation and concentrations likely have something to do with a change to the ecology and biogeochemistry brought about by the physical dynamics alluded to by Chen and McKinley. Further, we have adapted Figure 2a in addition to moving our previous supplementary figure on oxygen saturation dynamics to the main text (see comments below). Now, Figure 2a shows oxygen saturation instead of oxygen concentration, removing effects from changes in solubility due to temperature changes from seasonal mixing and isopycnal subduction. We also added grey contours of isopycnal surfaces at 0.1 kg/m^3 resolution in addition to the average mixed layer depth. We hope this helps to illustrate along-isopycnal changes in oxygen saturation over the seasonal cycle. This figure now shows the increase in oxygen saturation after deep mixing in the narrow band of isopycnals directly beneath the mixed layer, as opposed to increases in total oxygen concentration that occur across a broader range of depths.

Also, the question is really about primary production rates versus rates of heterotrophy. While it does not seem that these rates were measured on the cruise in question, primary production rates are measured at BATS. Are the rates higher or lower in the SOM below the mixed layer compared to mixed layer? Bacterial production is also measured at BATS, or at least was in Steinberg et al 2001. Please incorporate these data into your arguments. Are rates of primary production greater than bacterial production? This information is critical to understanding what is going on in the SOM.

We agree with this suggestion and have examined the historical rates. However, the BATS record generally missed the SOM because of sampling at standardized depths that do not focus on dynamic water column structure (this is common in many long term data sets). What we can see, based on our analysis, is that during the stratified periods in the few times (~7/112 experiments we analyzed) where the sampling did overlap the instances of primary production (PP) and bacterial production (BP) are positively correlated with high oxygen saturation (Figure S2). However, given the paucity of rate

data aligned to SOMs in the historical record, we cannot statistically conclude whether PP or BP rates were higher/lower. This analysis, combined with our flow cytometry data showing a lowered ratio of heterotrophic bacteria to *Pro* at the SOM (Lines 245-250_, could suggest higher primary productivity relative to secondary productivity. However, as stated later in the review, cell counts don't necessarily equate to higher/lower rates of PP or BP. Yet, we will include the new rate data in supplemental and discuss our analysis within the main text, but given that the survey does not directly sample the features of interest for this study we do not include it as a main figure. We have included the analysis of the existing BATS rate data as a supplemental figure (Figure S2), briefly report the findings in the abstract (lines 61-62), results (lines 185-197), and added methodology (line 473-480).

Detailed comments

I received this paper to review from Nature Communications. My impression is that Nature Communications has a requirement for separate Results and Discussion sections (though I defer to the editor on this). This paper needs Results and Discussion headings.

We have adjusted the manuscript structure to include a combined Results and Discussion section header following the Introduction, and have kept the sub-sections throughout the results and discussion with italicized subheadings. We made the ultimate decision to keep the results and discussion combined based on the collective feedback that there is too much discussion within each sub-section to just call it "results". However, we also defer to the editor for the final decision on this point.

Lines 56-61: ipolonies, numbers of infected *Prochlorococcus*, were left out of this sentence. However, they are the smoking gun!!! Please add in.

Addressed. We have now included the observation of elevated infection rates of *Pro* in the abstract (line 65).

line 87: the references are for the phototrophic populations and there are no citations for the oxygen concentrations. I guess Chen and McKinley 2019 should be cited here.

The appropriate references have been added for the oxygen reference (Chen and McKinley 2019, Steinburg et al. 2001). Line 82.

Line 100: Could this be elaborated on? What exactly is still elusive? The paragraph kind of makes it sound solved already. There is some room here to elaborate.

The elusive part is the specificity of how microorganisms/the microbial foodweb ("biology), potentially contribute to formation of SOMs. Specifically, dominant phototrophs such as *Prochlorococcus* playing a role as well as the often overlooked viruses that infect these populations (previous studies only counted particles). At present, no analysis of data on microbial activity exists to specifically ask questions about the potential role of microbes in SOMs. We included a few sentences highlighting the lack of microbial ecology studies at the SOM which hopefully helps the reader understand what we feel is still elusive – Lines 96-100.

Line 102: use a comma instead of a dash

The original sentence was reworded and no longer includes a dash (lines 102-103).

Line 155: List the range of years that these CTD casts come from: (1988-2019). Readers get to the methods last in these papers and so need some help.

The range of years has been replaced "since 1988 to 2019". Line 155.

Lines 171-172: You could quantify this. How much more oxygen was there in August compared to when the water subducted?

We have included some results regarding this comment in lines 169-184 which reference new main figure 3. Specifically, we report seasonal oxygen dynamics of isopycnals directly below the mixed layer in new figure 3 and found that oxygen saturation is ~95% during deep mixing in March compared to ~108% in August on average.

Line 179: Order by depth: upper mixed layer, SOM, DCM

The order of the depths has been fixed. Line 204

Line 183: The abbreviations are getting excessive here. I think you can write out the word surface. Also, just write out base of the mixed layer. Excessive abbreviation are a barrier to reader understanding.

The abbreviations for upper mixed layer, surface, and base of the mixed layer have been removed and replaced with the actual word throughout the text to avoid reader confusion. However, abbreviations are kept in the figures as abbreviated due to size constraints, but are defined in the figure legends.

Line 197 etc: using metagenomic data, (Fuchsman and Hays, 2023; Fuchsman et al., 2023) clearly show the same Prochlorococcus ecotypes in the mixed layer and right below the mixed, but different ecotypes dominating the DCM in the subtropical North Atlantic. Also some good information about cyanophage dynamics.

The relevant literature suggested has been added to the results (lines 228) and discussion regarding cyanophage dynamics (lines 372-374).

Line 210: were. Past tense

Tense has been fixed (line 232)

Line 219: I am not sure what sub-surface mixed layer peaks means. I mean, I figured it out, but maybe explain more clearly.

The sentence has been reworded and simplified to just “below the mixed layer” (line 242-244)

Line 228-230: Really you want primary production rates and bacterial production rates for these analyses. Not cell numbers.

This is a valid point and we have attempted to address this by assessing the BATS historical primary and secondary production rates as suggested (see explanation above). We still include the cell number results (lines 248-253) because we have this data for the cruise and to contextualize the findings relative to the rate measurements discussed in lines previously.

Line 254 and lines 268-274: There is usually an ammonium peak at the DCM or right below the peak of the DCM. So this is easily explained. You can see such an ammonium peak in (Widner et al., 2018) in the Pacific, but there must be similar data from the Atlantic. I believe it is a typical feature in the ocean.

Woodward and Rees (2001) Deep Sea Research Part II shows the ammonium peak in

the Atlantic. There is also Rees et al 2006 Deep Sea Research Part II.
I think you can assume this feature is also present at BATS.

We start by noting that these regions are difficult to directly compare biogeochemically with respect to the nutrients that are constraining productivity (as that constraint would change nutrient accumulation and consumption). We secondarily comment that accumulations of nutrients (in this case ammonium) can result from many processes - not the least of which is the biological supply of these nutrients through the actions of the microbial loop or viral shunt.

In the case of the Sargasso, it is widely held that P-availability limits productivity (see Lomas et al. 2010 Biogeosciences). This suggests then that the N supply could be (generally) sufficient. But the question arises as to the mechanism of that supply. In this study we are providing data that support the theory that viruses may be playing an important role in the resupply of DON to heterotrophs which then respire that material and seem to return the ammonium to Prochlorococcus. We do not mean to argue that there is no deep supply of ammonium. We point out that the intensity of amt transcription does not simply directly increase with depth as Prochlorococcus communities get closer to that deep ammonium supply, but is instead peaking in the SOM.

Line 259: lower than the mixed layer? Or the DCM?

Lowest at the SOM compared to all other measured depths (DCM, surface, base of mixed layer). The wording has been adjusted to clarify what we compared (line 265-268)

Line 274: That is really too bad!

Agreed

Line 311: Also seen in (Fuchsman and Hays, 2023)

The observations in Fuchsman and Hays, 2023 have been included in Lines 378-379

Line 344: viral isolates

Fixed – line 358

Line 432: lol! I am sure you can find a paper than uses these criteria, which would be

better than a webpage. Probably one of Dave Karl's tomes. Anyway, fix this.
Karl et al 2021 says "depths of upper mixed layer based on the density criterion of de Boyer Montégut et al. (2004)," so maybe track down that one.

Got it! de Boyer Montégut et al 2004 has been added for the mixed layer depth method.
(Line 460)

Line 435: List the shallower depths first

Done (line 463)

Line 454-455: You already said how you calculated the mixed layer above.

The explanation has been removed from the section and the section above has been referenced for BATS historical data processing. (lines 480-490)

Line 478-480: Do you have a time estimate of how long it took to get the RNA from the Niskin bottles to the freezer? This is critical for determining quality.

It takes us <1h (often just 15min) to get the RNA from the Niskins to the freezer. We set up a pump system where samples were filtered directly from the collection bottles and through the Sterivex simultaneously (6 samples at a time), and only 3L were filtered per sample. For the 8a/8p timepoints, there were 8 samples per cast. For the 12a/4a/12p/4p samples, there were only 4 samples collected per cast. The time from collection to freezing has been added in the Methods in Line 515.

Line 644: This sequence data needs to be submitted to NCBI SRA. [NG1] The people at JGI will do this for you. But it needs to be done. And you need to add a supplemental table with all of the accession numbers and numbers of good quality reads.

The libraries have been uploaded to NCBI SRA and accession numbers added to Supplementary Table 4. The data regarding good quality reads has also been added to Supplementary Table 4. A sentence has been added to Line 683 to point the readers to the information in the table.

Line 644: The BATS bottle data should also be specifically listed out here.

Sentences (Line 470-497) to the Methods (*BATS CTD data processing* section) have been added to list out the specific BATS bottle and rate data used.

Line 658: I also have a DOE CSP grant and was told to also include the award doi in publication. I think you should also be acknowledging the staff that helped with sequencing and sequence analysis. Please look at the directions related to your award.

Project staff that helped with the CSP has now been included in the Acknowledgments section (line 697)

Please put the supplemental figures and their figure captions into one combined word document!!!!!!!!!!!!!!!!!!!!!!

Supplemental figures and their captions will be submitted in one combined word document.

Since this is now a Nature Communications Document, the authors could take a few of the many supplemental figures and make them main document figures. I would recommend moving Figure S2 (SOM data from BATS) and Figure S4 Prochlorococcus ecotypes. It clearly shows that the SOM is more similar to the mixed layer than to the DCM.

As suggested, we moved Figure S2 to main Figure 3. Figure S4 has been included in additional panels in main Figure 4. Renumbering of the Figures has been done accordingly.

Bibliography

Fuchsman, C.A., Garcia Prieto, D., Hays, M.D., and Cram, J.A. (2023) Associations between picocyanobacterial ecotypes and cyanophage host genes across ocean basins and depth. PeerJ 11: e14924.

Fuchsman, C.A. and Hays, M.D. (2023) Increased cyanophage infection at the bottom of the euphotic zone, especially in the fall. Environ Microbiol 25: 3349–3363.

Widner, B., Mordy, C.W., and Mulholland, M.R. (2018) Cyanate distribution and uptake above and within the Eastern Tropical South Pacific oxygen deficient zone. Limnol Oceanogr 63: S177–S192.

Reviewer #2 (Remarks to the Author):

Gilbert et al. is a study examining the role of viruses in the formation of SOMs in water at the BATS site. Using geochemistry, transcriptional activity, and cell counts over timeseries they conclude that viral predation enhances oxygen production and NPP. The manuscript is well written, and the approaches are adequately support the conclusions. There are no major issues with this study.

We thank the reviewer for their feedback and have addressed all specific issues in the responses below.

Specific issues:

Ln 85 - Suggested re-wording to remove the repetition of "by" in this sentence.

The first "by" has been replaced with via to reduce repetition. (Line 80)

Ln 260 - Suggested change to: that were significantly

The wording was changed using this suggestion (Line 279)

Ln 272 - Are the strains in the SOM part of this group with constitutively high amt transcription?

It is possible that we captured an ecotype at the SOM that may constitutively express amt, however we see a mixture of different ecotypes (mainly, HLI and HLII, although HLII is more abundant relatively based on transcripts, see new Figure 4). The strains referenced in the literature w/ constitutively high amt transcription are MED4 (HLI) and MIT9313 (LLIV). The LLIV ecotype was not detected in abundance at the SOM, and HLI is not as abundant at the SOM compared to HLII. We have included the ecotypes referenced from the literature cited in the sentence regarding the constitutive expression (Line 292-293).

Ln 287 - Are there references that could be cited here to back up this assumption that transporters are induced in the presence of their substrates?

Refs 31 and 32 include evidence of this, however we removed the sentence in this section to make this paragraph more concise.

Ln 432 – missing citation

The citation for the MLD definition has now been included (de Boyer Montégut et al 2004) - Line 455

Ln 435 - This doesn't matter for understanding but this sentence may read more easily if all the layers are listed from either minimum to maximum (i.e. 5-40 m, 100-120 m, 50-60 m) or max-min.

Reviewer 1 had the same comment, and so the depths are now ordered from minimum to maximum (Lines 458)

Ln 551 - Which Illumina instrument was used?

NovaSeq S4 (same as the metaTs) - model used has been added to Line 582

Ln 558 - The recommendation for clustering on large-scales (>1,000s of sequences like you have here), according to the ClusterGenomes github page, is to use anicalc and aniclust in the CheckV distribution instead of ClusterGenomes.

I would s anicalc and aniclust from CheckV and verify the ClusterGenomes clustering accurately clustered the large number of viral sequences from this study.

Based on the suggestion, the dataset was run using anicalc/aniclust (same identity thresholds) from CheckV to compare to the ClusterGenomes results originally reported in the manuscript. We found that ClusterGenomes dereplicated those to 4921 vOTUs (1365 were > 10kb), aniclust/anicalc dereplicated to 4977 vOTUs (1368 were > 10kb). The only difference in the vOTUs were the 56 that ClusterGenomes did not identify. Further, ClusterGenomes, which uses nucmer from MUMmer package, is a global aligner which is better for accuracy. Whereas anicalc/aniclust uses megablast which is for fast local alignments and good enough for small genomes like viruses – hence preferred by the community for ease of use and efficiency. Ultimately, both tools use the same calculations post alignment hence the same vOTUs identified by both tools; and while differences are expected, the vOTUs identified by ClusterGenomes are more accurate than those identified by anicalc/aniclust.

Reviewer #3 (Remarks to the Author):

I co-reviewed this manuscript with one of the reviewers who provided the listed reports. This is part of the Nature Communications initiative to facilitate training in peer review

and to provide appropriate recognition for Early Career Researchers who co-review manuscripts.

REVIEWERS' COMMENTS

Reviewer #1 (Remarks to the Author):

This paper is much improved. As I said previously, this work is noteworthy for actually linking viral infection and biogeochemistry. The results are excellent. I list out small problems below.

Details

Line 68 Abstract: hmmm.... Usually people say that increased transcription of transporters indicates limitation. Here you say the opposite. I might edit or cut this sentence.

--The main text deals with this issue very well. Perhaps, in the abstract, expand the sentence to say elevated amt combined with lower ntcA transcripts indicate a response to the presence of increased ammonium.

Text has been changed to read "Cruise data also showed *Prochlorococcus* nitrogen metabolism transcripts consistent with increased responsiveness to remineralization activity by heterotrophs." We kept this text less specific to still fit in the abstract without leading to the awkward reading you point out.

Line 98: Change comma to period.

Done

Line 151: "The SOM during this study" – at first I thought you were referring to the SOM in the paper described in the previous sentence. Perhaps "our study"

Done

Line 169: You work in scientific papers should be in the past tense. Identified

Line 173: Indicated

Line 188: correlated

Line 190: measure is used awkwardly here

Tenses changed and line 190 changed to "Furthermore, the experiments that have the highest oxygen saturation and most closely resemble a SOM also have the highest primary and secondary production rates in the BATS climatology " to remove the awkward use of 'measure'

Line 227-228: The sentence "A peak in *Prochlorococcus* HLI has been observed below the mixed layer in the Mediterranean Sea." Did that occur during a SOM? Yes, it did. See Haro-Moreno et al 2018 "Fine metagenomic profile of the Mediterranean..." Table 1.

Text changed from "observed below the mixed layer" to "observed in a SOM". The reference (Haro-Moreno et al 2018) has also been included at the end of the sentence.

Line 277-278: Diel cycling is reduced at the DCM in general. See Vislova et al 2019. “Diel oscillation of microbial gene transcripts declines with depth in oligotrophic ocean waters”

We agree with this comment - our point is that instead of monotonically decreasing with depth in our study, the diel cycle in amt transcripts appears to have a higher daily max in the SOM than both the DCM and the surface.

Lines 311-313: This is nice.

Thank you.

Lines 394-396: And primary production!!!! Add in.

Primary production added to list.

Line 411: but Prochlorococcus always increases amt expression at dusk??? This is normal. Vislova et al 2019.

We agree with this point, with this sentence we just wanted to use an additional piece of evidence (the beam attenuation) to argue that the diel cycle in ammonium transporters is coupled directly to community growth.

Line 682: Raw meta-transcriptome and virome reads should be deposited at NCBI SRA. This is a community standard.

The NCBI SRA Ids for have been included in the Data Availability Section as well as Supplementary Table 4 (lines 695 – 696).

It is possible that Table 1 should be supplemental just do to the fact that it is so big.

Table 1 is now supplementary table 3 due to size constraints.

Figure 1c—red green color blind people can't read this one.

Fig 4bc—same

I know that red green color-blind people seem theoretical, but I have two in my lab. It is more common than you think.

We have adjusted the color schemes in Figure 1c and Fig 4bc to account for red/green color blindness.

Reviewer #2 (Remarks to the Author):

I feel all my comments were addressed adequately.

Reviewer #3 (Remarks to the Author):
